

# Taxonomic monograph of *Oxygyne* (Thismiaceae), rare achlorophyllous mycoheterotrophs with strongly disjunct distribution

Martin Cheek[1], Hirokazu Tsukaya[2], Paula J. Rudall[1] and Kenji Suetsugu[3]

[1] Science, Royal Botanic Gardens, Kew, Richmond, Surrey, UK
[2] Department of Biological Sciences, Graduate School of Science, University of Tokyo, Tokyo, Japan
[3] Department of Biology, Graduate School of Science, Kobe University, Kobe, Japan

Corresponding author
Martin Cheek, m.cheek@kew.org

## ABSTRACT

*Oxygyne* Schltr. (Thismiaceae) is a rare and little-known genus of achlorophyllous mycoheterotrophic perennial herbs with one of the most remarkable distributions of all angiosperm plant genera globally, being disjunct between Japan and West–Central Africa. Each species is known only from a single location, and in most cases from a single specimen. This monographic study names, describes and maps two new species, *Oxygyne duncanii* Cheek from cloud forest in SW Region Cameroon and *O. frankei* Cheek from gallery forest in the Central African Republic, representing the first new *Oxygyne* species described from Africa in 112 years, and raising the number of described *Oxygyne* species from four to six. *Oxygyne duncanii* is remarkable for sharing more morphological characters with two of the three Japanese species (*O. hyodoi* C.Abe & Akasawa, *O. shinzatoi* (H. Ohashi) Tsukaya) than with the geographically much closer type species of the genus, *O. triandra* from Mt Cameroon. Based mainly on herbarium specimens and field observations made in Cameroon and Japan during a series of botanical surveys, we provide descriptions, synonymy, mapping and extinction risk assessments for each species of *Oxygyne*, together with keys to the genera of Thismiaceae and the species of *Oxygyne*. The subterranean structures of African *Oxygyne* are described for the first time, and found to be consistent with those of the Japanese species. We review and reject an earlier proposal that the Japanese species should be segregated from the African species as a separate genus, *Saionia* Hatus. The only character that separates the two disjunct species groups is now flower colour: blue or partly-blue in the Japanese species compared with orange-brown in the African species. Studies of the pollination biology and mycorrhizal partners of *Oxygyne* are still lacking. Two of the six species, *O. triandra* Schltr. and *O. hyodoi*, appear to be extinct, and the remaining four are assessed as Critically Endangered using the IUCN 2012 criteria. To avoid further extinction, an urgent requirement is for conservation management of the surviving species in the wild. Since few achlorophyllous mycoheterotrophs have been successfully cultivated from seed to maturity, ex situ conservation will not be viable for these species and protection in the wild is the only viable option. While natural habitat survives, further botanical surveys could yet reveal additional new species between Central Africa and Japan.

# INTRODUCTION

*Oxygyne* Schltr. (Thismiaceae–Dioscoreales) is a rare and little-known genus of achlorophyllous mycoheterotrophic perennial herbs. It possesses one of the most remarkable distributions of all angiosperm plant genera globally, being disjunct between Japan and West–Central Africa (Fig. 1). Each species is known only from a single location, and in most cases from a single specimen.

During a series of botanical surveys carried out since 1991 in Cameroon to inform conservation management (*Cheek & Cable, 1997*), identification of collected specimens has resulted in the discovery of many species that are new to science, including a new species of *Oxygyne* Schltr. (Thismiaceae), previously designated as *Oxygyne* sp. nov. (*Thomas & Cheek, 1992*; *Cheek et al., 1996*; *Cheek & Ndam, 1996*; *Cheek & Williams, 2000*; *Cheek, 2006*; *Sainge et al., 2010*). Here, this specimen is formally named as *Oxygyne duncanii* Cheek, representing the first new African *Oxygyne* described for 112 years. This new species is remarkable for characters that are otherwise unknown in the sole previously known African species (*O. triandra* Schltr.) but which do occur in two of the three Japanese species (*O. hyodoi* C. Abe & Akasawa, *O. shinzatoi* (H. Ohashi) Tsukaya). In the course of finalizing this paper, material of a third African *Oxygyne* came to light, formally named here as *O. frankei* Cheek. Its morphological affinities are with the type species, *O. triandra*. In this monographic study, we present a detailed survey of all described species of *Oxygyne,* including mapping and extinction risk assessments for each species, together with a species key.

Although some earlier authors (*Jonker, 1938*; *Maas-van de Kamer, 1998*) placed *Thismia* and its allies (including *Oxygyne*) as a tribe Thismieae within the family Burmanniaceae sensu lato, recent molecular phylogenetic data (*Merckx et al., 2006*) strongly indicate that Thismiaceae are best placed as a separate family. These relatively well-sampled analyses place Thismiaceae as sister to Taccaceae, in a different subclade of Dioscoreales from Burmanniaceae sensu stricto. Furthermore, the two families are separated from each other by numerous morphological characters, as follows:

Burmanniaceae: Perianth long-cylindrical, only the upper portion detaching in fruit; stigma and stamens positioned at throat of tube; stamens three, sessile, anther cells lateral on broad central connective; stigmas often cupular; annulus absent; ovary septal nectaries present; ovary 3-locular, or 1-locular.

Thismiaceae: perianth cup-like or campanulate, rarely shortly cylindrical, detaching in fruit at base of perianth tube; stigmas inserted in basal $\frac{1}{4}$ to $\frac{1}{2}$ of perianth tube; stamens inserted at mouth of tube, or deep inside tube; stamens six (three in *Oxygyne*), on long filaments; anther cells collateral, not separated by connective; stigmas lobed; annulus present at mouth of tube or deep inside (except *Tiputinia* and *Haplothismia*); ovary septal nectaries absent; ovary 1-locular.

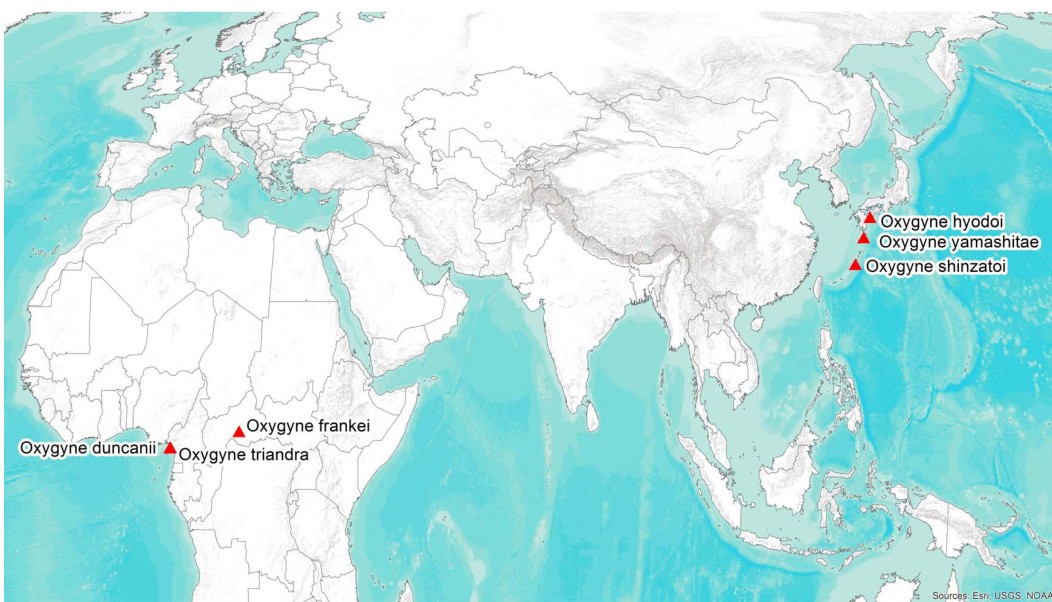

**Figure 1** **Global distribution of the species of the African–Japanese genus *Oxygyne*; drawn by George Gosline.**

*Oxygyne* differs from other Thismiaceae in some respects, notably in possessing only three stamens; it has therefore been included in either Burmanniaceae or Thismiaceae. However, in his monograph of Burmanniaceae, *Jonker (1938)* considered *Oxygyne* to be closer to Thismieae (today's Thismiaceae), stating ''... This important genus, of which unfortunately only one species and one specimen is collected .... shows the normal flower construction of the Thismieae: perianth urceolate to campanulate, style short and thick with three stigmas, ovary with three free stalked placentas (as in *Thismia*), but has only three stamens, while all other genera of Thismieae possess six stamens. A one-celled ovary and three stamens are the characteristics of the tribe Apterieae, but that tribe always shows quite another construction of the flower. The genus *Oxygyne* is therefore a link between the somewhat isolated *Thismieae* and the other tribes of the Burmanniaceae. For this reason I do not agree with Hutchinson's view to class the Thismieae as a separate family ...'' (*Jonker, 1938*: 47). However, more recent molecular data have supported Hutchinson's conclusion (e.g., *Yokoyama et al., 2008*; see below and Fig. 2).

## MATERIAL AND METHODS

The electronic version of this article in Portable Document Format (PDF) will represent a published work according to the International Code of Nomenclature for algae, fungi, and plants (ICN), and hence the new names contained in the electronic version are effectively published under that Code from the electronic edition alone. In addition, new names contained in this work which have been issued with identifiers by IPNI will eventually be made available to the Global Names Index. The IPNI LSIDs can be resolved and the associated information viewed through any standard web browser by appending the LSID

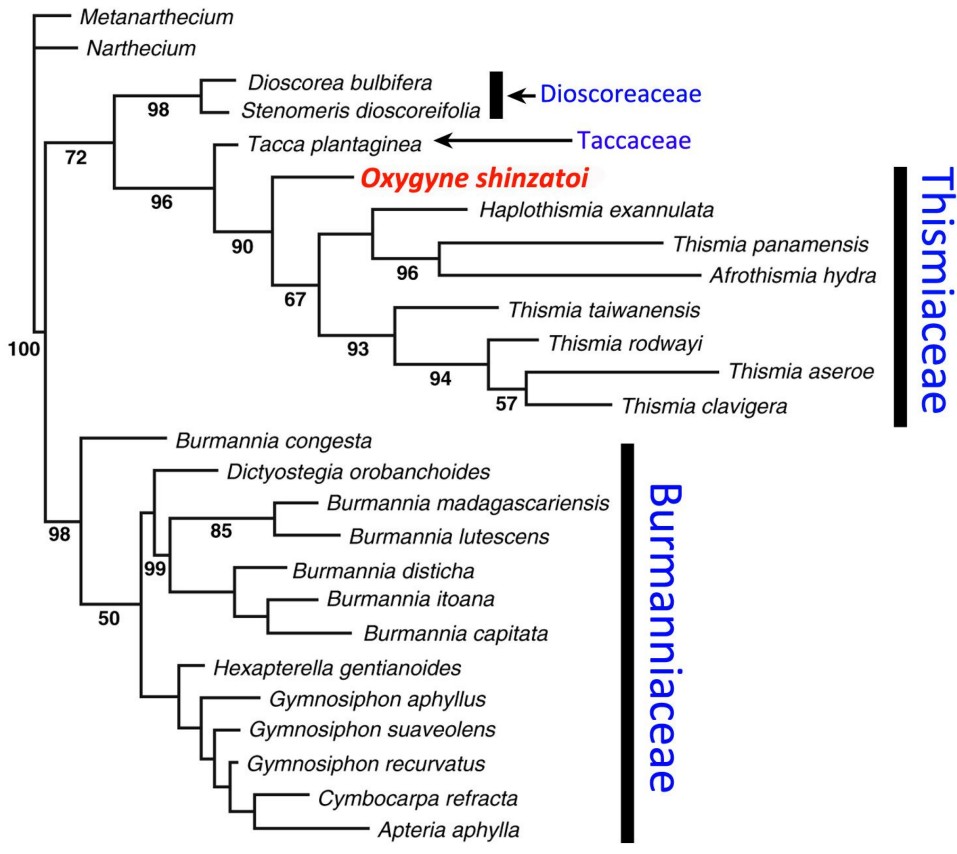

**Figure 2** Phylogenetic placement of *Oxygyne* based on phylogeny in *Yokoyama et al. (2008)*.

contained in this publication to the prefix "http://ipni.org/". The online version of this work is archived and available from the following digital repositories: PeerJ, PubMed Central, and CLOCKSS.

Nomenclatural changes were made according to the Code (*McNeill et al., 2012*). Names of species and authors follow *IPNI (2012)*. Regarding the Cameroonian species, this study is partly based on herbarium specimens and field observations made in Cameroon during a series of botanical surveys beginning in 1991. These surveys were mainly led by the first author. So far, they have resulted in 52,450 specimens being studied at K and YA, of which 37,850 were newly collected, the data stored on the Kew Cameroon specimen Access database (Gosline, p. 11 in *Cheek et al., 2004*). The methodology used is reported in *Cheek & Cable (1997)*. The top set of specimens was initially deposited at SCA, and later YA, duplicates being sent to K. The fieldwork was approved by the Institutional Review Board of the Royal Botanic Gardens, Kew entitled the Overseas Fieldwork Committee (OFC). The most recent OFC approval is numbered 807. The most recent invitation to effect research on the flora and vegetation of Cameroon has the reference number 050/IRAD/DG/CRRA-NK/SSRB-HN/09/2016. It is issued under the terms of the five year Memorandum of Collaboration between Institute for Research in

Agricultural Development (IRAD)-Herbier National du Cameroun and Royal Botanic Gardens, Kew signed 5th Sept 2014.

The morphological species concept was followed in defining species (each species being separated from its congeners by several, usually qualitative, morphological disjunctions), and the overall morphology of species was described and illustrated following standard botanical procedures as documented in *Davis & Heywood (1963)*. All specimens cited have been seen by one or more of the authors unless indicated 'n.v.' Herbarium citations follow Index Herbariorum (*Thiers, 2018*) and binomial authorities *IPNI (2012)*. Material of the suspected new species was compared morphologically with material of all other species of *Oxygyne* globally, which is contained at mainly at B, BM, FU, K, P, SCA TI and TNS.

Conservation assessments were either taken from the recent literature (see citations) or made using the categories and criteria of *IUCN (2012)*. Herbarium material was examined with a Leica Wild M8 dissecting binocular microscope. This was fitted with an eyepiece graticule measuring in units of 0.025 mm at maximum magnification. The drawing was made with the same equipment using Leica 308700 camera lucida attachment. Google Earth was used to georeference specimen points for mapping purposes. Specimen points were located on imagery from the metadata provided and latitude and longitude read-off from Google Earth.

For light microscope observations of root anatomy (Fig. 3), fixed material was sectioned using standard methods of Paraplast embedding and serial sectioning (6–12.5 um thickness) with a Reichert Jung 2040 rotary microtome. Sections were stained in safranin and Alcian blue and mounted in DPX. Photomicrographs were taken using a Leitz Diaplan photomicroscope (Leitz, Germany).

## TAXONOMIC TREATMENT

Thismiaceae J. Agardh, nom. cons. (*Agardh, 1858*: 99)
Type genus: *Thismia* Griff. (*Griffith, 1845*: 221)
Achlorophyllous mycoheterotrophs; for family description see *Stevens (2001 onwards)*.
Five genera, c. 55 species, mostly rare and narrowly endemic; overall distribution widely scattered, mostly in the tropical and subtropical forests of South America, Africa and Southeast Asia but extending into the temperate zone; in Africa with highest concentration of species in Cameroon (*Sainge, Chuyong & Peterson, 2017*).

### Key to genera of Thismiaceae

1.  Staminal filaments inserted at the mouth of the perianth tube, erect and exserted at base, then arching inwards, the anthers held at the level of the tube mouth, or just inside .................................................................................................................... 2

    Staminal filaments inserted deep inside the perianth tube, or (*Thismia*) in the throat, but filaments never erect or arching near the mouth, anthers held inside the tube .................................................................................................................... 4

2.  Stamens 6, annulus absent; anthers lacking a flat elliptic connective from which the thecae depend .................................................................................................................... 3

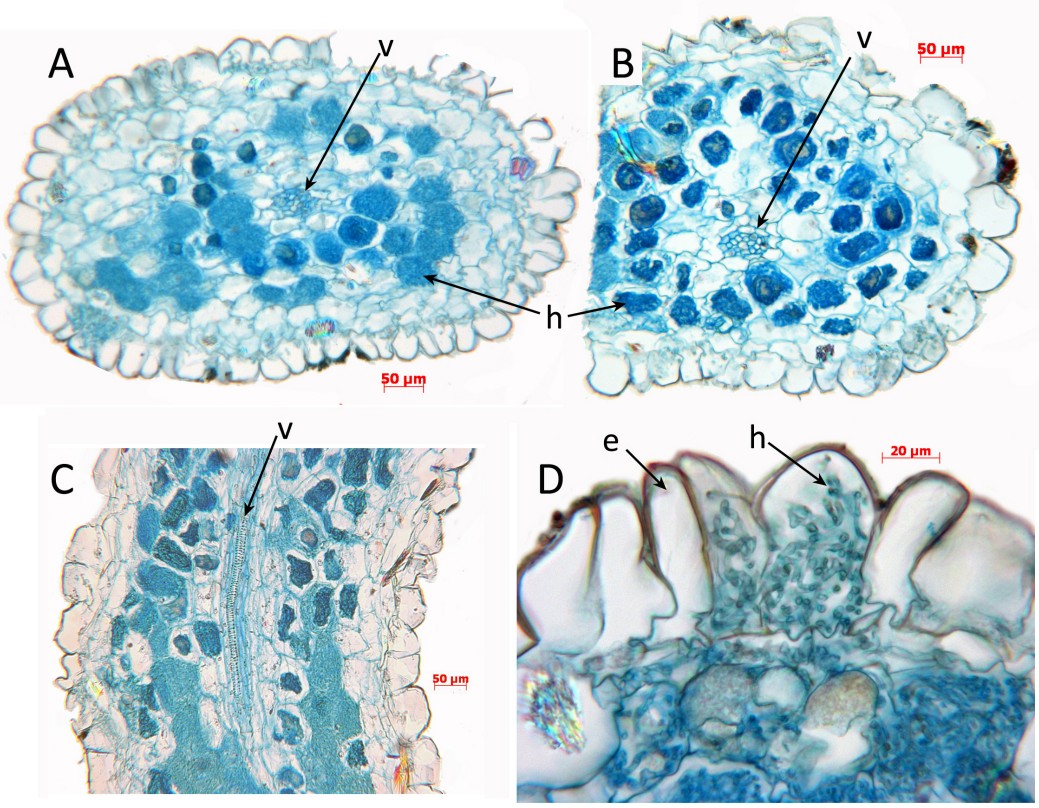

**Figure 3** *Oxygyne duncanii,* **root anatomy.** (A, B) Transverse sections through roots, showing central vascular strand and coils of enlarged hyphae in the cortex. (C) Longitudinal section of root showing central vascular strand and coils of hyphae in the cortex. (D) Part of root transverse section showing detail of epidermis and cortex. Labels: e, epidermis; h, hyphae; v, vascular strand. Scale bars = 100 μm. All photos by Paula Rudall.

Stamens 3, annulus lamellae present; anther thecae depending from a flat elliptic connective. Africa and Japan .......................................................................... *Oxygyne*

3. Inflorescences aerial, with multiple branches and flowers; India ........ *Haplothismia*

  Inflorescences at ground level, usually 1-flowered; Ecuador ..................... *Tiputinia*

4. Annulus inserted deep inside the perianth tube; stamens inserted below the annulus; anthers adnate to stigma; rhizomes with clusters of spherical tubers. Continental Africa ................................................................................. *Afrothismia*

  Annulus inserted at mouth of perianth tube; staminal filaments adjacent to annulus; anthers free, not touching stigma; rhizomes tuberous or not, but never with clusters of spherical bulbils. Americas, Asia-Australasia ..............................*Thismia*

*Oxygyne* Schltr. (*Schlechter, 1906*)

Type species: *Oxygyne triandra* Schltr.

Achlorophyllous mycoheterotrophs, glabrous, flowers held just above the substrate surface, the whole plant about 25(–50) mm high from tip of root to top of flower. Roots white (1–)3–6, vermiform, radiating from the base of the rhizome, rarely branched and then articulated at the junctions, c.1 mm diameter, lacking root hairs, surface papillate.

Stems rhizomatous, 1–3 per plant, white, ±erect, unbranched, terete, 1–1.5 mm diam; scale-leaves inserted spirally, 2–3 on the upper half of the rhizome, translucent, oblong or lanceolate, obtuse.

Inflorescences 1–2(–5)-flowered, bracts 3, in one whorl, equal, erect, subtending the flower, oblong-triangular, slightly convex, becoming accrescent and valvate around the fruit. Flowers bisexual, actinomorphic, blue, blue–green, blue–white or orange–brown, erect, 5–10 mm wide, perianth tube campanulate, lacking ridges; perianth lobes 6(–8), the three outer and three inner lobes equal and at anthesis appearing as if in a single whorl, inserted at rim of perianth tube, patent (or slightly ascending or erect, but possibly only as an artefact of preservation), narrowly triangular in outline and flattened, or filiform, variously ornamented or not.

Androecium with 3(–4) stamens inserted at mouth of perianth tube, opposite the outer perianth lobes. Filaments ascending at base, then arching down into the perianth tube, terete or flattened, and with or without appendages; anthers basifixed, extrorse, thecae ±elliptic, collateral, longitudinally dehiscent, inserted on the lower surface of a circular-elliptic connective, free. Annulus (or taenia) inserted immediately below the stamens, occluding the mouth of the perianth tube apart from a central pore and/or 3 apertures through which the stamens are inserted; composed of separate, connivent or united subquadrangular lamellae, each opposite the inner perianth lobes; lamellae absent (*O. triandra, O.shinzatoi, O.duncanii*) or present (*O. yamashitae, O.hyodoi*) opposite the outer perianth lobes; lamellae with appendages or central aperture or not.

Ovary white, campanulate, unilocular, placentae 3, either (*O. triandra*) "about three times the length of the placenta-stalks" (*Jonker, 1938*: 261) or (*O.duncanii*) clavate, the stipe attached distally near the junction with the floor of the perianth tube, placental heads dilated, with multiple ovules. Style-stigma $\frac{1}{4}$ to $\frac{1}{2}$ the length of the perianth tube, style terete, style-head/stigma with 3-angled or lobed head, and with three lateral lobes, the lobes erect acute, entire or bifurcate, or globose (*O. triandra* and *O. frankei* with three subulate lobes only, without lateral lobes).

Fruit almost enveloped by the three membranous, valvate, accrescent bracts; perianth falling above the base of the perianth tube. Fruiting ovary translucent-white, narrowly cylindrical-campanulate, apex truncate (floor of the perianth tube) developing an aperture by deliquescence, with a ragged margin. Seeds numerous, ovoid, lacking wings or elaiosomes, epidermal cells convex (*O. frankei*). Six species.

**Phenology:** Flowering in (July-)October.

**Distribution & habitat**: SW Region of Cameroon, Central African Republic, and southern Ryuku islands and southern mainland of Japan. Known from evergreen forest close to the ocean in mountainous areas with rainfall >2,500 mm p.a. (excepting *O. frankei*); 180–1,300 m alt.

**Etymology:** meaning sharp female organs, referring to the acute, prong-like styles of the type species.

**Local name**: various in Japan, none known in Africa.

**Conservation:** all species are Critically Endangered according to IUCN 2012 criteria, except for *O. triandra* and *O. hyodoi* which seem to be globally extinct.

Pollination has not been intensively investigated either in the Cameroon species (*O. duncanii,* this paper) or the Japanese species (e.g., *O. yamashitae*: *Yahara & Tsukaya, 2008*). However, unidentified small dipterans were recorded as the floral visitors of *O. yamashitae* (Suetsugu, pers. obs., 2015), as in *Afrothismia* (M Cheek, pers. obs., 2007). No scent has been detected in the flowers of the Cameroon species *O. duncanii* (this paper) or the Japanese species (e.g., *O. yamashitae*: *Yahara & Tsukaya, 2008*).

Seed dispersal has not been reported in *Oxygyne*. Seeds are probably dispersed by the splash-cup mechanism as reported for *Thismia* by *Maas-van de Kamer (1998)*. This inference is supported by the reported presentation of the fruits, which are held erect and have terminal openings. However, subsequent or alternative dispersal of seeds by ants, as occurs in *Sciaphila* (*Suetsugu, Shitara & Yamawo, 2017*) or even by camel crickets, as occurs in other Japanese mycoheterotrophs (*Suetsugu, 2017*) cannot be entirely ruled out. Elaiosomes, which occur in *Afrothismia* (*Cheek, 2009*), are not seen in the only species of *Oxygyne* in which seeds have been reported, namely *O. frankei* (this paper).

Root anatomy was examined here in *O. duncanii* (Fig. 3). Roots show the "typical" monocot structure (e.g., *Kauff, Rudall & Conran, 2000*), with a single-layered epidermis enclosing a cortex and central vascular strand. The cortex is parenchymatous, and the outermost layer of the cortex is not differentiated into a dimorphic hypodermis, as in some other monocots. Many of the cortical cells contain enlarged coiled fungal hyphae, which are also visible in some epidermal cells. Similar structures are also reported in other Thismiaceae (*Afrothismia*) and Burmanniaceae (*Imhof et al., 2013*).

No species of *Oxygyne* has yet been studied for fungal associates, but we speculate that the fungal partners of *Oxygyne* are likely to be arbuscular mycorrhizal (AM) fungi of the phylum Glomeromycota, as in other Thismiaceae. Most notably, *Afrothismia* has highly specific interactions with *Rhizophagus* fungi (*Franke et al., 2006*; *Merckx & Bidartondo, 2008*; *Gomes et al., 2016*; *Merckx et al., 2017*).

Cultivation of *Oxygyne* has not yet been reported and is unlikely to be successful since it has been rarely achieved for any other obligate mycoheterotrophic vascular plant (but see *Yagame et al., 2007*).

Chromosome numbers have been reported as $2n = 18$ in *O. shinzatoi* (*Tsukaya, Yokota & Okada, 2007*).

Two of the three African species occur or occurred in the Cross-Sanaga River interval, including western Cameroon. This region is the most species-diverse per degree square documented in tropical Africa; it includes several Pleistocene refuge areas (*Cheek et al., 2001*) and has been documented as a centre of plant diversity (*Cheek et al., 1994*). The Japanese species also occur (or occurred) in the species-diverse forests of Japan (*Yahara & Tsukaya, 2008*; *Suetsugu & Fukunaga, 2016*).

In terms of phylogenetic placement within Dioscoreales, molecular analysis using 18S rDNA (*Yokoyama et al., 2008*) placed *O. shinzatoi* within the *Thismia* clade (Thismieae or Thismiaceae), most likely as sister to all other species of Thismiaceae (see Fig. 2, also *Merckx & Smets, 2014*). Morphological analysis (*Merckx & Smets, 2014*) found that *Thismia* is paraphyletic with respect to *Tiputinia, Haplothismia,* and *Oxygyne*, though with low branch support. Our observations of comparative morphology indicate a close affinity

for *Oxygyne* with *Tiputinia* and *Haplothismia*, which share staminal filaments inserted at the mouth of the perianth tube which are initially erect, and later arch into the mouth of the tube. They also share six identical perianth lobes in which there is no differentiation between the outer and inner perianth whorl. It is possible that the lamellae that form the annulus in *Oxygyne* are homologous with the stamens opposite the inner perianth whorl in *Tiputinia* and *Haplothismia*. The globose structures at the base of the staminal filament in *O. duncanii* and the marginal teeth of the flattened staminal filaments of *O. yamashitae* could be homologous with similar structures in *Tiputinia foetida*. All species of *Oxygyne* share with the single species of *Haplothismia* anthers in which the two linear, collateral anther thecae depend from the midline of one face of a flat elliptic connective.

In this monograph, under the species entries below, we discuss taxonomic affinities based on morphology, but we choose not to erect any formal infrageneric groupings since so many gaps remain in our knowledge of the taxa recognized.

### Identification key to species of *Oxygyne*

1. Flowers blue–white or blue–green. Japan .................................................................. 2
   Flowers orange–brown or orange–brown and green. Central Africa ........................ 4
2. Staminal filaments strongly dorsiventrally flattened, subfoliose, margins deeply toothed; terminal style-stigma with three acute branches, style appendages each bifurcate, apices acute ........................................................................... **1. *O. yamashitae***
   Staminal filaments subcylindrical, margins entire; style-stigma and style appendages with apices clavate or capitate, rounded ...................................................................... 3
3. Perianth lobes filiform; base of lobes with callus-like cluster of round cells on the adaxial surfaces.............................................................................................**2. *O. shinzatoi***
   Perianth lobes narrowly triangular at base; base of lobes smooth .......... **3. *O. hyodoi***
4. Perianth lobes narrowly triangular; tube about as long as wide ......... **4. *O. duncanii***
   Perianth lobes filiform; tube about twice as long as broad ........................................ 5
5. Perianth lobes brown, c. 10 mm long; tube orange-brown. Cameroon ....................
   ................................................................................................................... **5. *O. triandra***
   Perianth lobes white, c. 5 mm long; tube dark green, base orange brown. C.A.R .......
   ...................................................................................................................... **6. *O. frankei***

**1. *Oxygyne yamashitae*** Yahara & Tsukaya (*Yahara & Tsukaya, 2008*:99; *Tsukaya, 2016*: 196)

Holotype: Japan, Kagoshima Prefecture, Yaku Island, along western branch of Futamata River, fl. 24 Oct. 2007, *K. Fuse, H. Yamashita & H. Ikeda* s.n. (Holotype FU! Herbarium specimen, isotypes TI! Herbarium specimens) (Figs. 1 and 4).

*Saionia yamashitae* (Yahara & Tsukaya) H. Ohashi (*Ohashi, 2015*:17). Homotypic synonym.

Small, achlorophyllous mycoheterotroph. Roots 1–5, creeping, c. 0.5–1 cm long. Stem simple or branched, erect, glabrous, less than 1 cm tall. Inflorescence stem short, c. 1 mm in diameter, racemose, 1- or 2(or 3)-flowered, bearing scale-like bracts, white. Bracts at base of flowers, three or more, 1 mm long, lanceolate, white. Flowers October, upright or inclined, pale blue, glabrous, c. 5 mm long, 5 mm in diameter; perianth united, 6(–8) lobed at apex, tube campanulate, c. 3 mm long; perianth lobes pale blue, narrowly triangular,

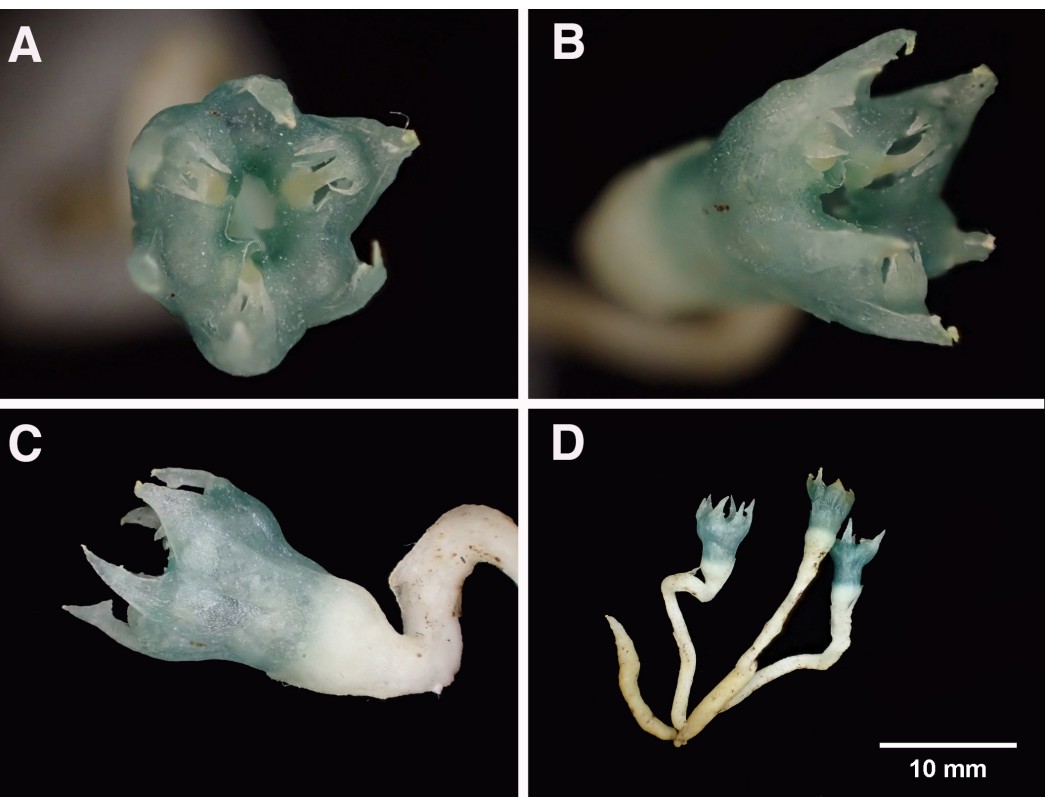

**Figure 4** *Oxygyne yamashitae.* (A) Flower, view from above showing three stamens, each with toothed filament; (B) three-quarter view of flower; (C) side view of flower, showing bracts; (D) whole plant, showing vermiform roots. All photos by Kenji Suetsugu.

elongated, c. 2 mm long, resembling a star; each perianth lobe having a lamella at throat; lamella convex, inflexed, united, forming an annular corona having a hole in centre; lamella of each outer perianth lobe having a slit in which a stamen is bent inward toward the stigma; lamella of each inner perianth 1 mm long, 0.7 mm wide, trapezoid, ended in a sharply bifurcated tip with two lobes 0.3 mm long. Stamens 3(–4), attached to base of outer perianth lobes; each filament inflexed, having two pairs of projections at base, bearing an anther at tip. Anthers pale yellow, bithecal, 0.5 mm long. Style 1, 1.3 mm long, thick, having 3 short, triangular stigmas, 0.25 mm long, 0.2 mm wide at base, surrounded with 3 dichotomous appendages. Stalks of appendages c. 0.15 mm long, bearing two unequal-sized branches elongated oppositely to each other along longitudinal axis of stigma; upper branches longer, pointed, c. 0.5 mm long, c. 0.1 mm in diameter; lower branches short, club-shaped, c. 0.15 mm long, c. 0.15 mm in diameter. Ovary c. 1 mm long, white. Ovules many (from *Tsukaya, 2016*).

**Phenology:** Flowering in October.

**Distribution & habitat**: known only from Yaku Island, formerly at two sites: (1) at Yudomari Forest (where now destroyed) and (2) western branch of Futamata River; 180–390 m altitude.

Humid evergreen forests near streams dominated by trees of *Machilus thunbergi* Siebold & Zucc., *Distylium racemosum* Siebold & Zucc., *Eurya japonica* Thunb., and *Ardisia sieboldii* Miq. The topography is flat, with c. 50% of the ground surface covered with three species of ferns: *Arachnoides amabilis* (Blume) Tindale, *Ctenitis subglandulosa* (Hance) Ching and *Diplazium domianum* (Mett.) Tardieu. Shrubs are *Damnacanthus indicus* Gaertn.f. and *Maesa japonica* (Thunb.) Moritzi ex Zoll. (*Yahara & Tsukaya, 2008*).

**Etymology:** commemorating Hiroaki Yamashita, photographer, of Yaku Island, who first noticed this species in Oct. 2000 and brought it to the attention of the botanist Yahara in Oct. 2006, after photographing it (*Yahara & Tsukaya, 2008*).

**Local name**: Yaku-no-hina-hoshi (*Yahara & Tsukaya, 2008*).

**Additional specimens**: none are known.

**Conservation:** *Oxygyne yamashitae* is not listed on iucn.redlist.org (accessed 30 April 2017), however, it is listed as Critically Endangered CR D on the Global Red List of Japanese Threatened Plants (http://www.kahaku.go.jp/ accessed 30 April 2017) which cites *Kato & Ebihara (2011)*.

*Yahara & Tsukaya (2008)* stated that of the two known populations of *O. yamashitae*, which was first discovered at Yudomari, one was destroyed by construction of a forest road. They expect that additional populations might yet be found in other forest in the area. However, so far none have been found (K Suetsugu, pers. comm., 2017). The second population at the Futamata River is in an area designated as National Forest, within which are allowed forest logging and construction, such as road construction, which destroyed the first population. Clearly, the only known surviving population is under threat. *Yahara & Tsukaya (2008)* provided detailed populational data. Yamashita noticed a plant of the first population in early October 2000, and at the second location in early October 2006, showing a photograph of it to Yahara soon after. A search by Yamashita and Yahara on 4th October 2007 failed to find any plants, but another search on 24th October 2007, by Yamashita and Fuse, recorded 11 flowering individuals, two of which were collected for study and preservation. Fuse, Yahara and Tsukaya revisited this site on 28 October 2007 and found more than 30 flowering individuals, four of which were collected.

*Oxygyne yamashitae* is perhaps the most distinct and morphologically isolated of all of the six known species of the genus: (1) While other species have the anthers inserted into the central aperture above the stigmas, or in slits between the lamellae (*O. duncanii*, *O. shinzatoi*) the anthers of *O. yamashitae* are unique in each being inserted into three separate apertures developed within the lamellae. (2) While in all the other species the stigmas are well included within the perianth, in *O. yamashitae* they are halfway along the tube. (3) The three lateral style appendages are entire in all species except *O. yamashitae*, where they are bifurcate. (4) The staminal filament in all species is more or less cylindrical (except *O. duncanii* and at the base in *O. frankei*) and lacks appendages, except in *O. yamashitae* where it is highly dorsiventrally flattened and bears two lateral pairs of long teeth, each far broader than the filament (Fig. 4); this latter character is also seen in *Tiputinia* and may be homologous.

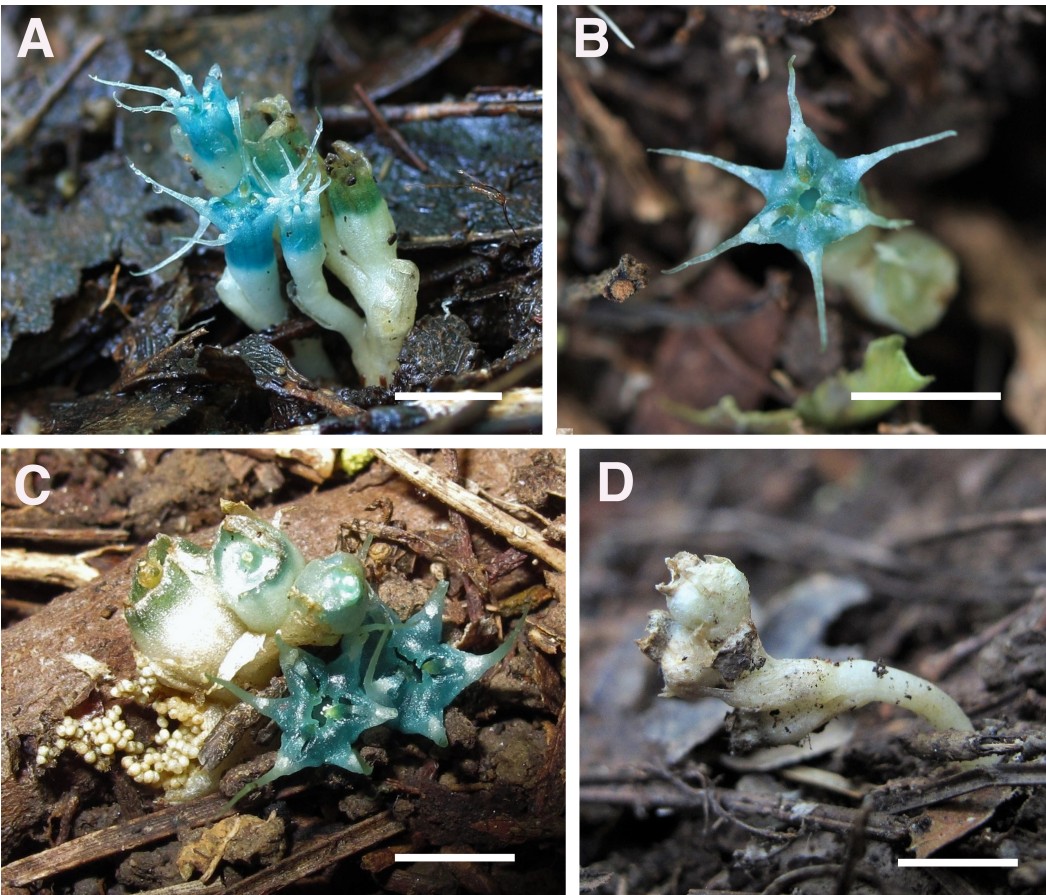

**Figure 5  Oxygyne shinzatoi.** (A) Cluster of flowers; (B) flower, from above; (C) flowers, fruit with shed seeds, *in situ*; (D) fruiting plant, side view. 24 Sept. 2008. Scale bar 5 mm. All photos by Tazuko Watanbe.

**2. *Oxygyne shinzatoi*** (H. Ohashi) Tsukaya (*Tsukaya, 2016*: 195)

Holotype: Japan, Ryukus Islands, Okinawa Prefecture, Okinawa Island, Yona, fl. 18 Sept 1974, *T. Shinzato* s.n. (RYU n.v. herbarium specimen) (Figs. 1 and 5).

*Saionia shinzatoi* H. Ohashi (*Ohashi, 2015*:116). Homotypic Synonym.

*Saionia shinzatoi* Hatus. (*Hatusima, 1975*: 909; *Hatusima & Nackejima, 1979*: 342, fig. 583; *Hatusima & Amano, 1994*: 326; *Okinawa Prefectural Government, 1996*: 111) nom. inval.

*Oxygyne shinzatoi* (Hatus. ex H. Ohashi) C. Abe & Akasawa (*Abe & Akasawa, 1989*:163) nom. illeg.

Perennial herbs, achlorophyllous, mycoheterotrophic. Roots creeping, c. 2–3 cm long. Stem simple or branched, erect, glabrous, 1–2.5 cm tall. Inflorescence racemose, (1 or) 2–5-flowered, bearing scale-like bracts, white, shortly pedunculate; peduncle c. 1.2 mm in diameter. Bracts at base of flowers, (3–)5 or more, to 5 mm long, ovate-lanceolate to lanceolate, white. Flowers c. 5 mm long, 9–12 mm in diameter, bluish verdigris. Perianth tube campanulate, c. 5 mm long, 3 mm in diameter, 6-lobed, glabrous, lobes pale blue (young stage) or white (older stage), narrowly triangular, elongate, c. 3–5 mm long, 0.6 mm

wide at base. Each perianth lobe with callus-like cluster of round cells on adaxial surface at base; callus cluster 0.5 mm long; with 3 trapezoid lamellae from inner perianth lobes, trapezoid lamellae c. 0.7 mm wide, with 3 or 4 shallow apical teeth, bluish verdigris, curved downward from throat of perianth tube, forming 3 narrow slits and a small opening at centre of throat. Perianth lobes and lamella bluish verdigris. Stamens 3, elongating from base of outer perianth lobes, filaments curved downward between trapezoid lobes on throat of perianth tube; anthers apical, pale yellow, bithecal, 0.7 mm long. Ovary c. 3 mm long, white. Style 1, 1 mm long, thick, bilateral, surrounded by 2 or 3 appendages, with 2 or 3 short unequally sized sickle shaped stigmas; stigmas less than c. 0.4 mm long, 0.2 mm wide. Tip of style appendages ball-like, rounded, c. 0.3 mm in diameter, c. 0.5 mm long including stalk. Ovules many, 0.3 mm long. Fruit fusiform, bluish, verdigris; seeds numerous (from *Tsukaya, 2016*).

**Phenology:** Flowering in September and October.

**Distribution & habitat:** only known from the type location, in forest of *Pinus luchuensis* Mayr and *Schima wallichii* (DC) Korth. subsp. *liukiuensis* (Nakai) Bloemb. (*Hatusima, 1975*), and *Schima superba* Gardner & Champ. (*Okinawa Prefectural Government, 1996*) and *Machilus thunbergiii* Siebold & Zucc.*, Camellia japonica* L., *C. sasanqua* Thunb., *Dendropanax trifidus* (Thunb.) Makino ex Hara, and *Schefflera heptaphylla* (L.)Frodin (*Tsukaya, 2016*).

**Etymology:** commemorating the collector of the first specimens, T. Shinzato.

**Local name:** Hoshizaki-shakujuyô (*Hatusima, 1975*); Hoshizaki shakujou (*Japanese Ministry of the Environment, 2000*); Nansei-shoto (*Ministry of the Environment, Japan, 2016*).

**Additional specimens:** all from same location as the type: 20 Sept. 1972 *T. Shinzato* s.n. (RYU n.v.); ibid. (now an experimental field site of Ryukyu University) 8 Oct. 2006, *H. Tsukaya s.n.* (TI!, TNS!); ) 8 Oct. 2006, M. Yokota s.n. (RYU n.v.).

**Conservation:** *Oxygyne shinzatoi* is listed as Critically Endangered CR (D) on both http://iucn.redlist.org (accessed 30 April 2017: see below), and on the Global Red List of Japanese Threatened Plants (http://www.kahaku.go.jp/ accessed 30 April 2017) which cites *Kato & Ebihara (2011)*.

We concur with the assessment of Critically Endangered assigned this species by *Okinawa Prefectural Government (1996)* and *Kato & Ebihara (2011)*, even though we have not seen supporting evidence of either threats to *O. shinzatoi* or number of mature individuals being <50, or any of the other data needed to evidence this category under another criterion using the *IUCN (2012)* standard. This is because the species is known from a single location and is rarely seen. It has been reported only in the years 1972, 1974, 2004, 2008 and 2011. In 1996 it was reported that it had not been seen for 20 years (*Okinawa Prefectural Government, 1996*). However, it is not clear whether the plants had been regularly searched for each year at the correct season. The type location is private land and so is not open to visitors (K Suetsugu, pers. comm., 2017).

*Ministry of the Environment, Japan (2016)* are credited with the entry for this species on http://www.iucnredlist.org/, which also maintains the species as Critically Endangered (CR D) since "only several individuals were found during a search in 2011". Its

habitat is stated to be inside a University Forest (specifically attached to the Faculty of Agriculture, University of the Ryukyus at Yona, Kunigusuju-son, fide *Okinawa Prefectural Government, 1996*) and well protected, though small-scale logging is ongoing. In order to strengthen the protection of the subtropical evergreen broadleaved forest, including the habitat of this species, a new National Park was designated (*Ministry of the Environment, Japan, 2016*). Yambaru National Park, located in the northern part of Okinawa Island was designated as the 33rd National Park in Japan on September 15, 2016 https://www.env.go.jp/en/nature/nps/park/yambaru/point/index.html.

*Oxygyne shinzatoi* is perhaps the most well-studied of all species of *Oxygyne*; it is the only species for which cytological (*Tsukaya, Yokota & Okada, 2007*) and molecular phylogenetic studies (*Yokoyama et al., 2008*) exist. It remained nomenclaturally invalidly published for many years since its first discovery in 1972, since two holotypes (contrary to the Code) were proposed by its discoverer Hatusima in the original publications where it was named *Saionia shinzatoi* (*Hatusima, 1975*; *Hatusima, 1976*). *Abe & Akasawa* (*1989*:163), evidently unaware of the invalidity, had transferred the species to *Oxygyne* where it continued to be invalid. *Ohashi (2015)* rectified the matter of validity by selecting a single holotype, indicating that he expected the correct name to be then *Oxygyne shinzatoi* (Hatus. ex Ohashi) C. Abe & Akasawa. However, since there is no retroactivity in this matter according to the Code, it was not until *Tsukaya (2016)* made the combination that the name was validly transferred to *Oxygyne*.

**3. *Oxygyne hyodoi*** C. Abe & Akasawa (*Abe & Akasawa, 1989*:161 fig.1; *Tsukaya, 2016*: 196)

Holotype: Japan, Shikoku, Ehime Prefecture, Minamiuwa-gun, Nishiumi-Cho, in evergreen forest, fl. 9 Oct 1988, *Syozi Hyodo* s.n. (TI! Herbarium specimen). (Figs. 1 and 6).

*Saionia hyodoi* (C. Abe & Akasawa) H. Ohashi (*Ohashi, 2015*:117). Homotypic Synonym. Perennial herbs, achlorophyllous, mycoheterotrophic. Roots creeping, c. 1–2 cm long, internodes fusiform, creamy white. Stem simple, erect, glabrous, 2–3 cm tall. Inflorescence racemose, shortly pedunculate, 1–3-flowered. Leaves scale-like membranaceous, c. 1.2–1.6 mm long, emerald green when fresh. Bracts at base of flowers c. 3 mm long. Flowers c. 4 mm long, 2 mm wide, emerald green. Perianth urceolate-campanulate, tube c. 3 mm long, 6-lobed, glabrous, lobes c. 2 mm long, 3–4 mm wide, triangular-semiorbiculate, apices filiform. Annulus forming a convex dome over the mouth of the perianth tube, with a central aperture into which the stamens descend. Stamens 3, from junction between annular taenia and perianth lobes, filaments curved downward. Style 1, thick, bilateral, c. 1.4 mm long, with 3 clavate lateral appendages on upper surface, stigmas 3, short (modified from *Tsukaya, 2016*).

**Phenology:** Flowering in October.

**Distribution & habitat**: known only from the understorey of evergreen forest at the type location, where it was recorded growing with *Burmannia liukiuensis* Hayata (*Abe & Akasawa, 1989*).

**Etymology**: commemorating Syozi Hyodo, member of the Nippon Fernist Club, collector of the type specimen and discoverer of the first known specimen of *Oxygyne hyodoi* (*Abe & Akasawa, 1989*).

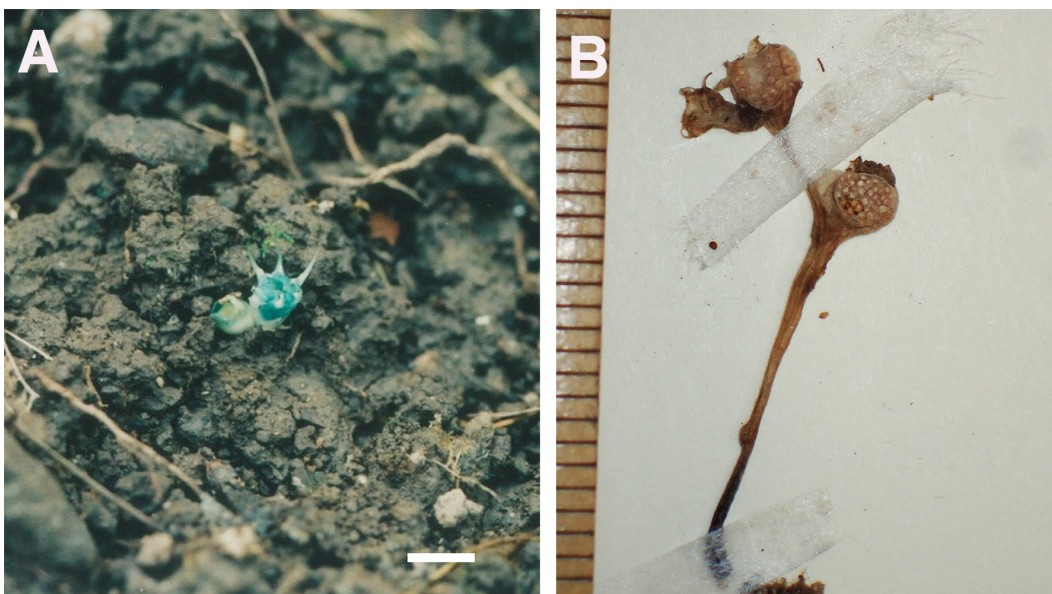

**Figure 6** *Oxygyne hyodoi.* (A) Live plant *in situ*, 1988 by Syozi Hyodo, scale bar 5 mm; (B) photo of type specimen *Hyodo s.n.* (TI), scale bar in mm, by Kenji Suetsugu.

**Local name**: Hina-no-bonbori (*Abe & Akasawa, 1989*; *Japanese Ministry of the Environment, 2000*).

**Additional specimens:** nil.

**Conservation:** *Oxygyne hyodoi* is not listed on http://iucn.redlist.org (accessed 30 April 2017), however, it is listed as Critically Endangered (CR D) on the Global Red List of Japanese Threatened Plants (http://www.kahaku.go.jp/ accessed 30 April 2017) which cites *Kato & Ebihara (2011)*.

However, we assess the species as extinct since "*Oxygyne hyodoi* has not been found again" (*Ohashi, 2015*: 115) despite repeated targeted searches for it at the type locality by botanists (e.g., "We could not find this species during our research in 2001" *Ehime Prefectural Government, 2003*) including those experienced in observing Japanese *Oxygyne* species (Suetsugu, pers. obs., 2017). Among these is the original discoverer of the species, Mr Hyodo, an enthusiastic field botanist who was reported to have returned to the original site every year for 15 years but has not succeeded in finding his species (S Gale, pers. comm., 2017). An immature specimen once tentatively considered to be this species, discovered near Kobe City, Hyogo Prefecture (*Kobayashi & Kobayashi, 1993*), is now considered most likely a *Thismia*. In any case, natural habitat at that site has been lost to a development project for an industrial park (*Hiraoka Environmental Science Laboratory, 2001*).

Drawings in the protologue of the flower in three dimensions clearly show that the stamens are inserted into the central aperture of the annulus dome, and that in fruit only the basal rim of the perianth tube walls, and its base, together with the style, remain.

The local name signifies "lantern of a lady doll", probably in reference to the shape of the perianth, according to S Yasuda (pers. comm., 2005).

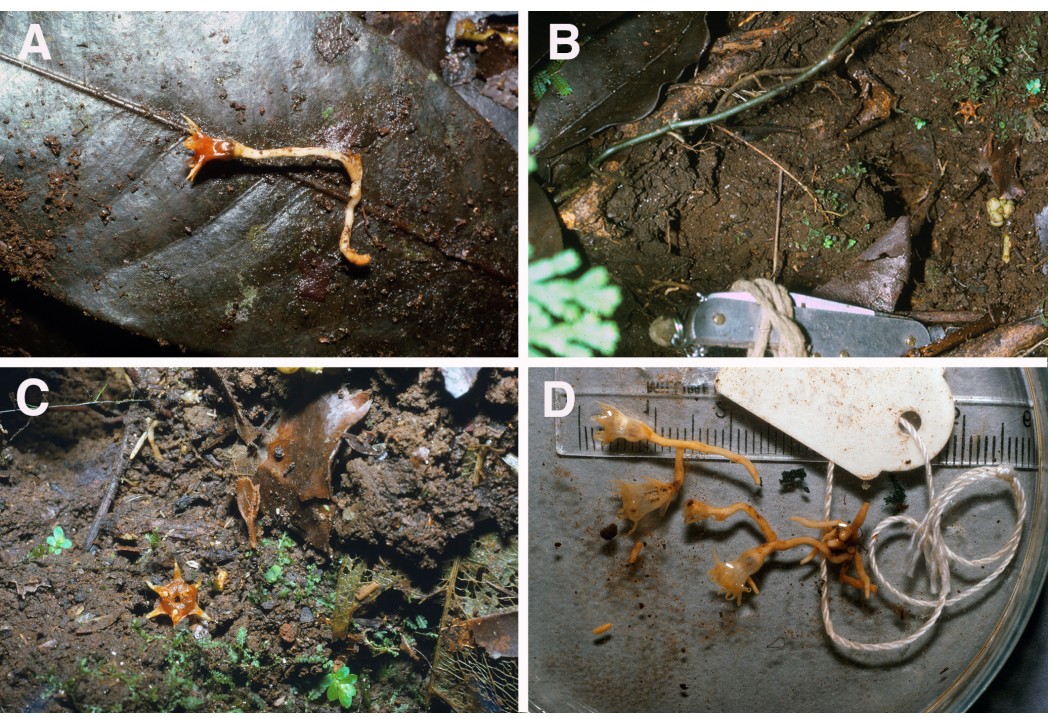

**Figure 7 _Oxygyne duncanii._** (A) Whole plant, side view; (B) in habitat (penknife for scale); (C) close-up of flower *in situ*; (D) preserved specimens in ethanol. All photos by Martin Cheek.

**4. _Oxygyne duncanii_** Cheek sp. nov.

Holotype: Cameroon, SW Region, saddle between Mt Cameroon and Mt Etinde, 1,300 m alt., fl. fr. 25 Oct. 1992, *Cheek* 3816 (holotypus SCA herbarium specimen!; isotypus K spirit specimen!) (Figs. 1, 7 and 8).

*Oxygyne sp. nov. Cheek & Ndam* (*1996*: 613, 614); *Cheek & Williams* (*2000*: 41); Cheek in *Cable & Cheek* (*1998*: lxv, 149).

**Affinities** (Diagnosis): differs from all other species of *Oxygyne* Schltr. in the presence of a pair of globose structures below the staminal filament insertion; the erect apiculus from the midpoint of the adaxial surface of the perianth lobes; and the stigma for which the central, distal stigmatic portion is 3-angled, not 3-lobed.

Achlorophyllous mycoheterotroph, glabrous, a single flower above the surface, the whole plant 25–30 mm high from tip of root to top of flower. Roots white 5–6, vermiform, radiating from the base of the rhizome, rarely branched and then articulated at the junctions, c.1 mm diameter, lacking root hairs, surface papillate. Stems rhizomatous, 1–3 per plant, white, ±erect, unbranched, terete, 1–1.5 mm diam; scale-leaves inserted spirally, 2–3 on the upper half of the rhizome, translucent, oblong or lanceolate, c. 2 × 0.6 mm, obtuse. Inflorescence 1-flowered, bracts 3, in one whorl, equal, erect, subtending the flower, hyaline, oblong-triangular, 2.5–3 × c.1.5 mm, slightly convex, becoming accrescent and valvate around the fruit. Flowers erect, c. 7 mm long, 9–10 mm wide, perianth tube brownish orange, black in 0.5 mm near ovary, glossy, campanulate, c. 3.5 mm long, c.2 mm

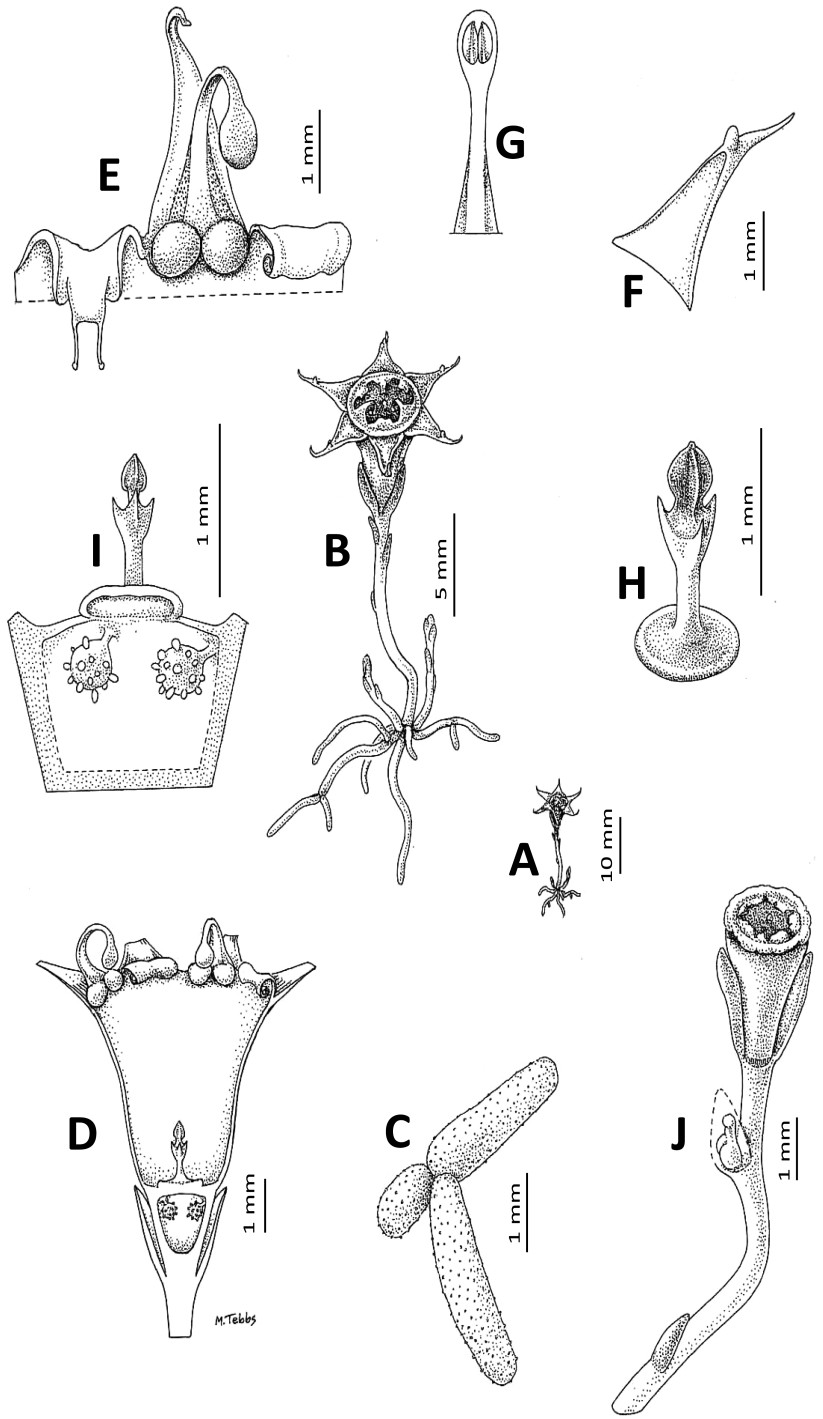

**Figure 8 *Oxygyne duncanii*.** (A) Habit; (B) habit, whole plant exposed; (C) root branch; (D) flower, sublongitudinal section; (E) detail of mouth of the opened-out perianth tube, viewed from inside showing the annulus lamellae tightly involute as in preservation (right) and unravelled (left); (F) perianth lobe, showing adaxial mucro; (G) part of stamen showing thecae on lower surface of the connective; (H) style-stigma, with disc; (I) ovary placentation, detail; (J) fruit. (A, B) from photo and from *Cheek* 3816, (C–H) from *Cheek* 3816. Drawn by Margaret Tebbs.

wide at the base, c. 4 mm wide at the apex, sometimes slightly constricted at the midpoint, even, lacking ridges; perianth lobes patent to slightly ascending, triangular, c. 3 × 1.5 mm, adaxial surface with margin raised in the proximal part, or the margins uniting 2 mm from the base, before an erect mucro 0.2–0.3 mm high, distal part of lobe extending a further 1 mm, subfiliform. Androecium with three fertile stamens inserted at junction of perianth lobes and tube, the filaments with an adaxial pair of globes at the base, each c. 0.6 mm diameter, inserted on the rim of the perianth tube, the filaments themselves dorsiventrally flattened, c. 3.3 mm long, c. 0.35 mm wide at base, c. 0.25 mm wide at apex, erect at base, then strongly incurved, anthers extrorse, thecae ± elliptic, collateral, c. 0.5 × 0.2 mm, inserted on the lower surface of a circular-elliptic connective c. 0.9 × 0.6 mm. Annulus inserted immediately below the stamens inside the mouth of the perianth tube, brownish-orange, incomplete, of 3 lamellae, each opposite the three sterile (inner) perianth lobes, in spirit material involute from apex and 1.5 mm wide; when unfurled with a proximal transversely rectangular portion, then transversely cylindrical, c. 0.5 mm diameter, c. 1 mm long, and a distal bifurcate portion c. 1 mm long, 0.5 mm wide, the apical sinus c. 0.5 mm deep, the two lobes filiform; *in vivo* the lamellae extend horizontally, seamlessly occluding the mouth of the perianth tube except for a depressed central circular pore c. 1 mm diameter and 3 anther slits between the lamellae.

Ovary white, campanulate, c. 1.5 mm long, c. 0.7 mm wide at base, 2 mm wide at apex, unilocular, placentae 3, attached distally near the junction with the perianth floor. Style-stigma 1.1–1.4 mm long, style terete, c. 0.5 × 0.2 mm style-head/stigma with a 3-angled head c. 0.5 × 0.5 mm and with three lateral lobes, the lobes erect, 0.3 –0.5 mm long, acute. Fruit (probably immature) translucent-white, narrowly cylindrical-campanulate, 3.5 × 2.5 mm, apex truncate and, pre-dehiscence, a central umbo (style remnant), the central part aperturate by deliquescence, with a ragged margin (perianth tube remnants); bracts almost enveloping the fruit, valvate, c. 3.2 × 2 mm long

**Phenology:** Flowering in October.

**Distribution & habitat:** North facing slope, c.10–20 m from bottom of a shady valley drained by a small (seasonal?) stream. Submontane evergreen forest dominated by *Garcinia* cf. *smeathmannii* (Planch. & Triana) N.Robson ex Spirel., with some *Pseudagrostistachys africana* Pax & K.Hoffm.*, Psychotria camptopus* Verdc. and *Cola* cf. *verticillata* Stapf ex A.Chev., herbs including *Plectranthus, Palisota mannii* C.B.Clarke*, Dracaena* spp., *Oplismenus hirtellus* (L.)P.Beauv.and *Selaginella* spp., *Impatiens etindensis* Cheek & Eb. Fisch. and *I. frithii* Cheek, *Renealmia* sp., *Pteris* spp.; 1,300–1,400 m alt.

**Etymology:** The epithet commemorates Duncan Thomas, the most prolific living collector of plant specimens in Cameroon, with whom the species was collected. He has striven to build the capacity of its botanists. It was Dr Thomas who supervised the forest plot enumeration during which the type specimen of *O. duncanii* came to light.

**Conservation:** This species is known from only four plants (three in flower, one in fruit) in one area of about two square metres, despite about four hours subsequent continuous searching by each of four botanists in the valley in which the discovery was made, and in the adjacent valley. No obvious threats were detected at the time of discovery, apart from a seasonally occupied hunters' camp nearby. Accordingly, using the criteria of *IUCN (2012)*,

*Oxygyne duncanii* is here formally assessed as CR D1, that is, Critically Endangered, since less than 50 mature individuals are known. The species was first assessed thus by Cheek (in *Cable & Cheek, 1998*: lxv), as *Oxygyne sp. nov.* However, since the species had not been formally published, it was not then possible for this assessment to be accepted by IUCN.

While it is not impossible that more localities exist for *O. duncanii*, it is worth noting that Mt Cameroon and Mt Etinde are relatively well-surveyed botanically by West African standards. During the period 1992–1994, a series of surveys at these locations produced 9,600 specimens, the basis for a checklist (Cheek in *Cable & Cheek, 1998*: xxi). Prior to this, at least 28 botanists had visited Mt Cameroon and collected specimens, beginning with Gustav Mann in 1861 (*Cable & Cheek, 1998*). While achlorophyllous mycoheterotrophs such as *Oxygyne* can easily be overlooked by the general collector, several mycoheterotroph specialists have worked on and around Mt Cameroon. These are documented in some detail under *Oxygyne triandra* (above). Most notably, Cameroon's most prolific discoverer of achlorophyllous mycoheterotrophs, Moses Sainge, is often based on Mt Cameroon a few km away at the University of Buea.

In contrast with *O. triandra,* which was recorded in a community with five other mycoheterotrophs (see under that species), no additional mycoheterotrophs were found in association with *O. duncanii,* despite searching over several hours by several botanists. Generally in Cameroon, mycoheterotrophs occur at relatively low altitudes, below 800 m, and it is unusual to find a species at such high altitude as this, at 1,300–1,400 m, in submontane forest. We conjecture that this submontane species evolved from a common ancestor with the lowland *O. triandra*, although morphologically it appears more similar to the Japanese species.

The horizontal separation of the type (and only) localities of *Oxygyne triandra* and *O. duncanii*, is about 12 km, the vertical separation about 1 km.

Mt Etinde, the subpeak of Mt Cameroon, has a different age and geology, and consequently, a different flora to that of Mt Cameroon, the main massif. Unlike Mt Cameroon, it is not volcanically active, but the result of geological uplift of plutonic rocks, and predates the larger mountain (*Cable & Cheek, 1998*). As a result, this region includes many rare species that are not found on the main massif, including several that are endemic to Etinde, such as *Coffea leonimontana* Stoff. (*Stoffelen, Robbrecht & Smets, 1997*) and *Impatiens etindensis* Cheek & Eb. Fisch. (*Cheek & Fischer, 1999*). It also contains species that are otherwise found only in the Bakossi Mts, c.60 km to the north of Mt Cameroon, including *Dracaena kupensis* Mwachala, Cheek, Eb.Fisch. & Muasya (*Mwachala et al., 2007*), *Deinbollia oreophila* Cheek (*Cheek & Etuge, 2009*) and *Impatiens frithii* Cheek (*Cheek & Csiba, 2002*).

No scent was detected from the flowers, nor pollinators observed during the hours of the afternoon when this plant was discovered and surveyed, and last seen (M Cheek, pers. obs., 2017).

In life the lamellae form a dome over the mouth of the perianth, with a central aperture and three inconspicuous radial anther slits (Fig. 7). It is possible that the filiform extensions of the lamellae with their capitate apices serve to join firmly the edges of the lamellae into a single structure by locking with each other. In the high rainfall habitat in which this and

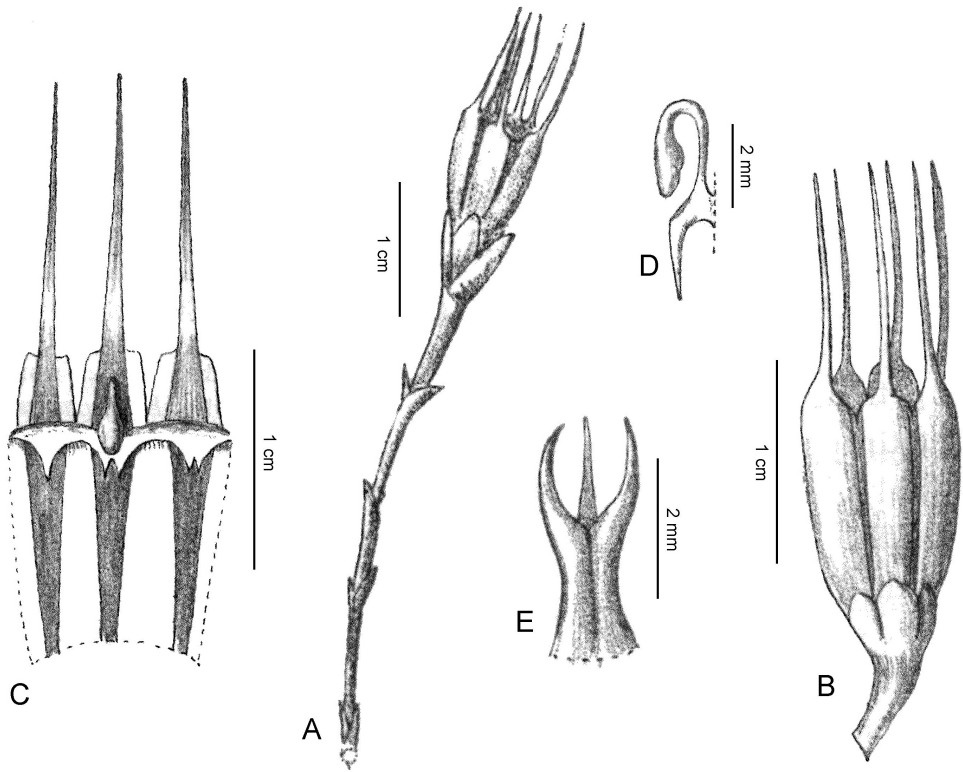

**Figure 9** *Oxygyne triandra.* (A) Habit; (B) flower; (C) perianth, part, opened, inner surface; (D) stamen and annulus in section, side view; (E) style-stigma. From *Schlechter (1906)*, drawn by Joseph Pohl (1864–1939).

most other species of the genus occur, the dome could serve to prevent raindrops from flooding the perianth tube, which would obstruct pollination and reproduction.

**5. *Oxygyne triandra*** Schltr. (*Schlechter, 1906*:140; *Jonker, 1938*: 47, 260–261; *Hepper, 1968*: 179)

Holotype: Cameroon, ''Urwald bei Moliwe'', Sept. 1905, *Schlechter* 15790 (holotype B, herbarium specimen). (Fig. 9)

Achlorophyllous mycoheterotroph, glabrous, 3.8 cm high. Underground parts unknown. Stems erect, unbranched, terete, 1–2 mm diameter; scale-leaves c. 10 per stem, inserted spirally, oblong-ovate, 1.25 × 1 mm, apex lacerate. Inflorescence 1-flowered, bracts 3, apparently in one whorl, equal, erect, subtending the flower, ovate, 8 mm long. Flower erect, c. 22 mm long, perianth tube dark brown, campanulate-cylindrical, c. 10 mm long, c.7 mm wide at the throat, longitudinally 6-pleated; perianth lobes orange-yellow, erect (possibly an artefact of preservation), 12 mm long, 1.55 mm wide, basal part oblong, apex narrowly caudate-elongate, highly acute. Androecium with stamens inserted at junction of corolla throat and base of outer perianth lobes, filaments terete, 0.1 mm wide, basally erect, above arched downwards; anthers elliptic, 1 × 0.75 mm, deflexed, apex rounded, lacking appendages; annulus of three lamellae, each opposite the inner perianth lobes, transversely oblong-ovate, c.1 × 0.75 mm, apex bifid, lobes 0.25 mm long, acuminate, each sometimes

shortly bifid. Style thick, 3-ribbed, stigma trifid, the branches erect, subulate. Ovary with three parietal placentae, each three times as long as the stalk (*Jonker, 1938*: 261). Fruits obovoid-truncate, c.4 mm long, apex of the persistent basal ring of the perianth and with the persistent style (*Jonker, 1938*: 261).

**Phenology:** Flowering in September.

**Distribution & habitat:** known only from the type specimen, which indicated "urwald bei Moliwe" (forest by Moliwe). Moliwe is in the geologically ancient eastern foothills (rising to 200–300 m asl.) of Mt Cameroon, in an area that was lowland evergreen rainforest, with annual rainfall of 3–4 m per annum (*Cheek, 1992*; *Cable & Cheek, 1998*). The ecology of the remnant of forest that remains, formerly known as Mabeta-Moliwe, now better known as Bimbia-Bonadikombe, was documented in detail in *Cheek (1992)*. A summary can be found in *Cable & Cheek (1998)*. Apart from *Oxygyne triandra,* numerous other narrowly endemic and threatened species are recorded from this forest, e.g., *Drypetes moliwensis* (*Cheek, Radcliffe-Smith & Faruk, 2000*), *Cola cecidifolia* Cheek (*Cheek, 2002*), *Salacia nigra* Cheek (*Gosline, Cheek & Kami, 2014*), *Psychotria moliwenis* Bridson & Cheek, *P. bimbiensis* Bridson & Cheek (*Cheek & Bridson, 2002*) and *Ancistrocladus grandiflorus* Cheek (*Cheek, 2000*).

**Etymology:** "triandra" signifies the three stamens, unique to this genus among thismiaceous genera.

**Local name:** none recorded.

**Additional specimens:** none are known.

**Conservation:** *Oxygyne triandra* was listed as Critically Endangered (CR A1c+2c) by Cheek (*Cable & Cheek, 1998*: lxv). This assessment was later published on the IUCN redlist website iucn.redlist.org where it is attributed to *Cheek & Cable (2000)*. The assessment of *Oxygyne triandra* was updated to CR B2ab(iii) +D by Cheek in *Onana & Cheek* (*2011*:358) where it was stated "the prospects of refinding this must be very low .....". *Oxygyne triandra* was given as an example of a species now thought to be globally extinct (*Cheek & Onana, 2011*:7). It has never been seen since it was first discovered in 1905, despite numerous targeted searches by multiple groups of expert achlorophyllous plant researchers at the correct season over many years (see below).

Although most of B was destroyed by allied bombing in 1943, many of the monocot specimens survived, including the type of *Oxygyne triandra* (http://ww2.bgbm.org/Herbarium/specimen.cfm?Barcode=B100277415).

Attempts to rediscover *Oxygyne triandra*

The Oxford University expedition to Mt Cameroon (July–Sept 1993) had the rediscovery of this species, together with *Afrothismia pachyantha* Schltr. and *A. winkleri* (Engl.) Schltr. as a key objective, but failed to find them in the Moliwe area or elsewhere in the lowlands of Mt Cameroon (*Baker, Hunt & Von Rege, 1995*). Targeted searches by the first author with teams of other botanists during 1992–2007 also failed to find *Oxygyne triandra*, initially at Moliwe and Mabeta–Moliwe, later in numerous other evergreen forest areas in SW Region Cameroon as part of comprehensive species surveys to support conservation

management. These surveys are documented by *Cheek et al. (2004)*, *Cheek, Harvey & Onana (2010)* and *Harvey et al. (2004)*, *Harvey, Tchiengue & Cheek (2010)*. Despite this, the surveys did discover several other achlorophyllous mycohetrotrophic species (*Cheek & Ndam, 1996*), including some new to science e.g., *Kupea martinetugei* Cheek & S. Williams (*Cheek, Williams & Etuge, 2003*; *Cheek, 2004a*)) and *Afrothismia amietii Cheek (2007)*. Further targeted searches for *Oxygyne triandra* by other research groups also failed to find the species, notably those over several years led by mycoheterotroph specialists Thassilo Franke (late 1990s to early 2000s) then by Vincent Merckx and Moses *Sainge et al. (2010)*. These expeditions also discovered new species of the closely related *Afrothismia* (*Franke, 2004*; *Franke, Sainge & Agerer, 2004*; *Merckx, 2008*). While new discoveries of mycoheterotrophs throughout Africa have extended the geographic range of the genus and since 2,000 more than tripled the number of species of *Afrothismia* (e.g., to Kenya: *Afrothismia baerae Cheek (2004b)*, to Tanzania: *Afrothismia mhoroana Cheek & Jannerup (2006)*, and Malawi *A. zambesiaca Cheek (2009)*, no further records of *Oxygyne* have been made, apart from *O. duncanii* and *O. frankei* (this paper).

 Analysis of Schlechter's specimens (*Cheek & Ndam, 1996*) reveals that he collected several other species of mycoheterotroph in a series consecutive with *Oxygyne triandra* (*Schlechter* 15790). This supports Schlechter's observation that his *Oxygyne triandra* occurred in a community with other Burmanniaceae sensu lato, and with *Sebaea,* just as Ridley had reported in the Malay Peninsula (*Schlechter, 1906*). Clustering together of mycoheterotrophs in one place is known in Africa and South America (*Schlechter, 1906*; *Cheek & Ndam, 1996*; *Cheek & Williams, 2000*). The consecutive Schlechter collections are:

| | |
|---|---|
| *Burmannia hexaptera* Schltr. | *Schlechter* 15785 & 15786 |
| *Burmannia congesta* (Wright) Jonker | *Schlechter* 15787 |
| *Afrothismia winkleri* (Engl.) Schltr. | *Schlechter* 15788 |
| *Afrothismia pachyantha* Schltr. | *Schlechter* 15789 |
| *Sebaea oligantha* Schinz | *Schlechter* (in obs.) |

 Examination of the type specimen in 2006 showed that the illustrations in the protologue (*Schlechter, 1906*) were misdrawn, in so far as (1) the stem is largely sheathed in scales, and (2) the annulus is not continuous, instead lamellae are present only opposite the inner (sterile) perianth lobes and are absent from the outer perianth lobes. The description above is based on that of the protologue modified by observations made of the type in 2006 by the first author. There are several discrepancies between the description in the protologue and that of *Jonker (1938)*. While Schlechter states that the plants were 1-flowered, and that fruit were absent, *Jonker (1938)* states that they are 3-flowered and describes the fruit.

 **6. *Oxygyne frankei* Cheek sp. nov.**
Holotype: Central African Republic, 40 km N Bambari, près Balimbwa, fl. fr. 25 July 1928, *Tisserant* 2623 (holotype BM000803697; isotypes P00319728, P00319729 images, all herbarium specimens) (Figs. 1 and 10).

**Affinities** (Diagnosis): differs from *Oxygyne triandra* Schltr. in the lamellae oblong, lacking lobes, in the perianth lobes white, 5–5.5 mm long, (not brown, 10 mm long).

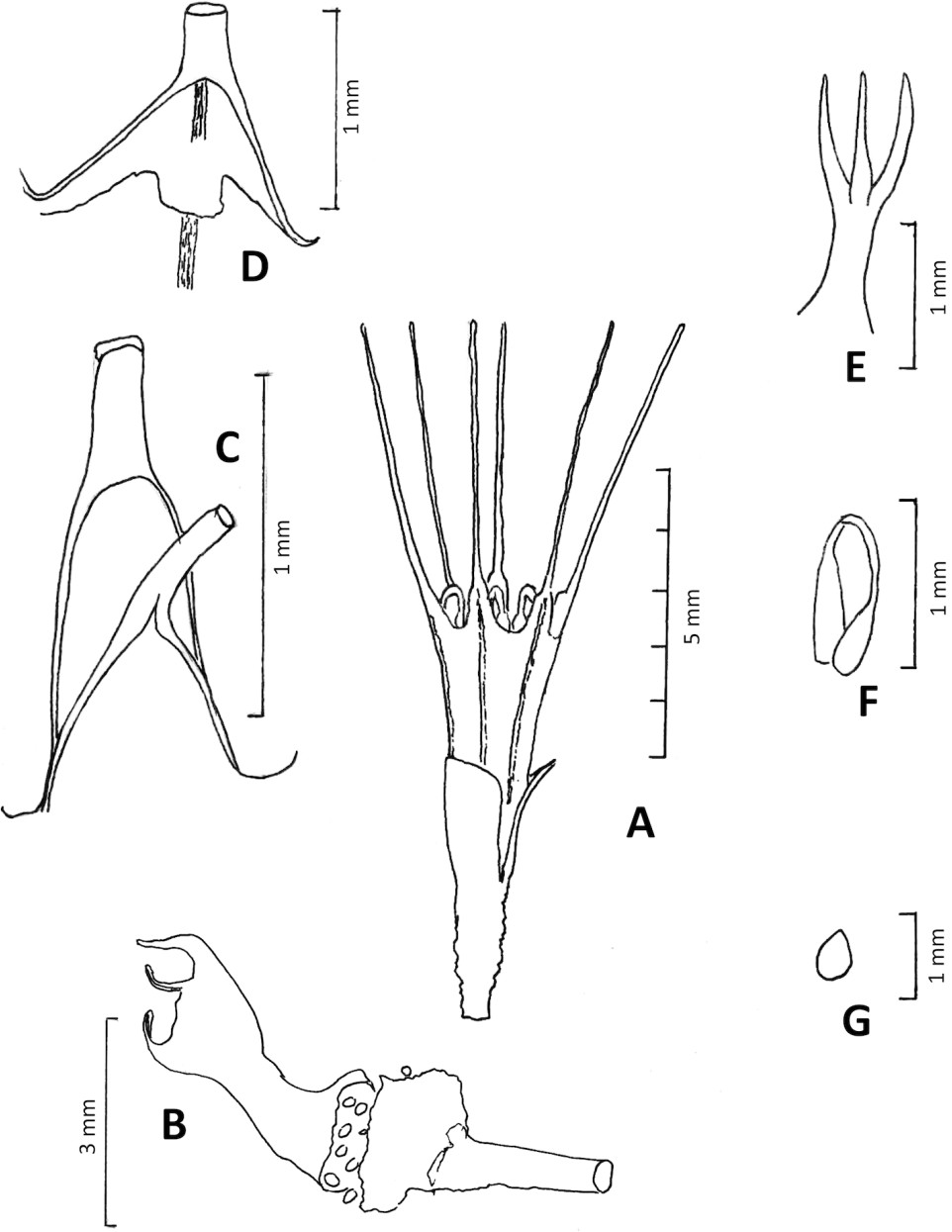

**Figure 10 Oxygyne frankei.** (A) Habit, flowering plant at anthesis; (B) habit, fruiting plant with ripe seeds; (C) base of outer perianth together with base of staminal filament; (D) base of inner perianth lobe, showing the lamellum at its base; (E) style and stigmas; (F) stamen (G) seed. (A–G) from *Tisserant* 2623. Drawn by Martin Cheek.

Achlorophyllous mycoheterotroph, glabrous, a single flower above the surface, the perianth tube 2–3 mm diameter. Roots not seen. Stems rhizomatous, white, ±erect, unbranched, terete, 0.5–0.75 mm diam; scale-leaves not seen. Inflorescence 1–(2-)flowered, bracts several, erect, subtending the flower, hyaline, oblong, c. 0.8 × 0.2 mm, apex obtuse or acuminate, slightly convex, becoming persistent around the fruit. Flowers erect, dark

green, base of the perianth orange-brown, tube campanulate, 3.5–5 mm long, (2–)3 mm wide, widening gradually from base to apex, sometimes slightly constricted at midpoint, lacking ridges, sometimes with 6 longitudinal dark lines corresponding with the perianth lobes. Perianth lobes white, erect, 5–5.5 mm long, proximal part oblong or triangular, 0.8–1.2 mm long, 0.6–1 mm wide, apex narrowing abruptly to the long, linear distal part, 0.1–0.2 mm wide, margins involute, apex obtuse. Stamens 3, opposite outer perianth lobes, inserted at junction of perianth lobe and tube, the filament c. 1.5 mm long, flattened, proximal 2/5 triangular, base erect, about as wide as perianth lobe base, narrowing to the strongly incurved, filamentous, 0.1(–0.2) mm wide distal part; anthers borne at same level as filament base, extrorse, thecae collateral, inserted on the lower surface of connective, elliptic c. 0.3 × 0.2 mm. Annulus of 3 lamellae, inconspicuous, lamellae inserted at the same level as the stamens at the base of the inner (sterile) perianth lobes, each lamellum transversely oblong to quadrate, 0.2 × 0.2 mm, appressed to perianth tube (possible artefact of preservation). Ovary ellipsoid-campanulate, c. 1.5 mm long, c. 1 mm wide, unilocular, placentae 3, attached distally near the junction with the perianth floor. Style-stigma 2 mm long, style terete, 1 mm long, style with three erect, subulate-filiform lobes 1 mm long, apices acute. Fruit ellipsoid, 3 × 2 mm, bracts enveloping the fruit, 3.5–4 × 1.52 mm long. Seeds ovoid, bright yellow-orange, 0.3–0.4 mm long, epidermal cells convex.

**Phenology:** Flowering and fruiting in late July.

**Distribution & habitat:** Habitat not well recorded, presumed to be gallery forest "Under the roots of Gramineae, places more or less damp" ("sous racines du graminées, endroit humides"); c.450 m asl (read from Google Earth).

**Etymology:** Named for Thassilo Franke, formerly active in research on C. African achlorophyllous mycoheterotrophs, who first discovered this species to be a new *Oxygyne.*

**Local name:** none recorded.

**Additional specimens:** none are known.

**Conservation:** *Oxygyne frankei* is here assessed as Critically Endangered (CR D) since less than 50 mature individuals are recorded. No immediate threats are visible on Google Earth. No road access and no settlements appear in the area. However, iron ore mining is reported as a possibility in the area. Extensive botanical surveys were carried out in C.A.R. since the 1980s by Mike Fay and David Harris, but no additional collections of Thismiaceae have been made (D Harris, pers. comm., 2017). However, no targeted searches have been made for this species, and since habitat in the type location appears to be intact, there is every possibility that the species survives there. Botanical surveys are not currently advisable in C.A.R. due to the security risk (https://www.gov.uk/foreign-travel-advice/central-african-republic visited 10 Dec. 2017).

**Notes:** The holotype was formerly in the private collection of Georges le Testu (1877–1967) who bequeathed it to BM in 1967. The main label appears in the hand of the collector, the Reverend Charles Tisserant. His identification is "*Cytinacée, Hydnora*". The first annotation is a label signed "L.D. Gómez-P." dated Sept. 1983, with the inscription "*Thismia intermedia* sp. nov. Holotypus." This name was not published and the writer has not been traced. The specimen was then annotated as *Oxygyne* sp. by T. Franke in 2002,

and then as *Oxygyne* sp. nov. by T. Franke in 2008. However, T. Franke is reported to have left botany since that time.

The discovery of *O. frankei* in C.A.R. was unexpected for several reasons. In contrast with the locations of all other species of the genus, the Bambari area (in fact the C.A.R. as a whole), is not renowned for its high species diversity or for the presence of narrowly endemic species. Unlike all other species of the genus, this species occurs far from the sea (1,280 km as measured with Google Earth), in an area where evergreen forest does not predominate, and where rainfall is <2 m p.a. Rainfall at Bambari is reported as 1.47 m p.a., mainly falling (months with >100 mm) from April–October (https://en.wikipedia.org/wiki/Bambari downloaded 10 Dec. 2017). The predominant vegetation type is woodland. However, at the type locality 40 km N of Bambari appears a dense network of evergreen gallery forest on a low, flat plateau. These forest regions seem to depend on shallow, branching drainage lines. Although rainfall is below the 2 m threshold usually needed to support evergreen forest, the gallery forest is likely to be sustained by water draining from the slightly higher woodland areas.

That such a relictual, point-endemic species as *Oxygyne frankei* should occur in gallery forest in a predominantly woodland area indicates that other such point-endemics might be found if survey effort was focused in such habitats.

The morphological affinities of *O. frankei* are clearly with *O. triandra* of Cameroon. The preserved specimens of both species (observations or images of live plants being absent apart from colour notes) share the following features: (1) erect very narrowly triangular to caudate perianth lobes; (2) dark brown lines (corresponding with the vascular supply) extending from the base of the perianth tube to the base of each perianth lobe; (3) reduced lamellae which are too small to occlude the perianth tube mouth; (4) a stigma with three subulate curved-erect lobes only, lacking lateral lobes. These features are shared with no other species of the genus, and may be worth recognizing at the level of subgenus. Should future molecular data support generic separation, these two species would comprise *Oxygyne* and the remaining species *O. duncanii* and the three Japanese species, *Saionia*.

*Oxygyne confusa* Bidault, Merckx & Byng (in *Christenhusz, Fay & Byng, 2018*: 55) was published online in February 2018 while this paper was being revised. Based on the same specimen *Tisserant 2623*, this name predates *Oxygyne frankei*. However, because *Christenhusz, Fay & Byng (2018)* contains nearly 4,000 controversial new combinations, the entire work is being considered for suppression by the International Association of Plant Taxonomy under the Code as an "Opera Utique Oppressa". Acceptance of the name *O. confusa* here would undermine the case for suppression by recognising that work. However should the case to suppress the work fail, *Oxygyne frankei* will be replaced by *Oxygyne confusa* because it has precedence.

## DISCUSSION

### Discovery and extinction

The specimen *Schlechter* 15790 was collected in forest near Moliwe (in the eastern foothills of Mt Cameroon on the Cameroonian coast of the Gulf of Guinea, Africa) in 1905, resulting

in the recognition and publication of *Oxygyne* Schltr. based on *Oxygyne triandra* Schltr. (*Schlechter, 1906*). This species is now believed to be extinct (*Cheek & Onana, 2011*, see under species account above).

The second collection of *Oxygyne* was in July 1928 in the Central African Republic (C.A.R.) by Charles Tisserant (*Tisserant* 2623) but it was misidentified as *Hydnora* Thunb. (now Hydnoraceae) and kept in a private collection until 1967. Consequently it was only recently discovered to be an *Oxygyne*, described and is newly named in this paper as *O. frankei* Cheek.

The second species of *Oxygyne* discovered was collected (*Shinzato* s.n. in 1972) in Japan's southern Ryukyu Islands, specifically Okinawa Island, and invalidly published as *Saionia shinzatoi* Hatus. (*Hatusima, 1975*; *Hatusima, 1976*). The transfer to *Oxygyne* as *O. shinzatoi* (H.Ohashi) Tsukaya was not valid until *Tsukaya (2016)*.

The third species of *Oxygyne* discovered was collected in Japan's Shikoku Ehime Prefecture (*Syozi Hyodo* s.n., 9 Oct 1988) and named as *Oxygyne hyodoi* C. Abe & Akasawa (*Abe & Akasawa, 1989*). This was the first application of the name *Oxygyne* to a Japanese species. *Oxygyne hyodoi* is now considered extinct (see species account).

In October 1992, a fourth species of *Oxygyne* was discovered, on Mt Etinde, a sub-peak of Mt Cameroon in Cameroon: *Cheek* 3816 (*Thomas & Cheek, 1992*; *Peguy et al., 2000*; *Cheek et al., 1996*; *Cable & Cheek, 1998*; *Cheek, 1996*; *Cheek, 1997*; *Cheek, 2006*; *Cheek & Ndam, 1996*; *Cheek & Williams, 2000*). It is formally described and newly named in this paper as *Oxygyne duncanii*. It has not been re-recorded since its discovery (*Sainge, Chuyong & Peterson, 2017*).

The fifth discovered species of *Oxygyne*, the third from Japan, was viewed by Hiroaki Yamashita on the island of Yaku, South of Kyushu 24 Oct 2000, and later collected (Fuse, Yamashita & Ikeda s.n., 2008) and published as *Oxygyne yamashitae Yahara & Tsukaya (2008)*.

In terms of distance, the remarkable Central African–Japan range disjunction of *Oxygyne* is rivalled only by a few other disjunct distributions reported within other fully mycoheterotrophic genera, including an Amphi-Pacific Northern Hemisphere disjunction in *Thismia* (*Merckx & Smets, 2014*). Among other achlorophyllous mycoheterophic disjuncts, *Geosiris* Baill. (Iridaceae) has one species in Madagascar, a second in the Comoro Islands and a third species recently discovered in northern Queensland, Australia (*Gray & Low, 2017*), and strong disjunctions occur within genera of Triuridaceae (*Rudall, Alves & Sajo, 2016*) and Corsiaceae (*Mennes et al., 2015*).

## One genus or two?

Despite the remarkably disjunct distribution of *Oxygyne*, our results support its retention as a single genus. *Hatusima (1976)* contended that *Oxygyne shinzatoi,* the first *Oxygyne* discovered in Japan, was generically distinct; he erected the name *Saionia* for his proposed new genus, with the subtribal name *Saioneae* Hatusima to accommodate it. He distinguished Saioneae "a subtrib. *Oxygynearum* differt perianthium fauce haud annulatum. Stamina in geniculum e lobi perianthii interiores inserta." Effectively, he stated that *Saionia* differs from *Oxygyne* in lacking a perianth annulus, and in the stamens being

geniculate and inserted at the inner base of the perianth lobes. However, in fact, both these features are present in the type species of *Oxygyne* (see below under *O. triandra*).

*Saionia* was for the most part not taken up by botanical authors, who continued to use *Oxygyne* for the Japanese species. For example, *Abe & Akasawa (1989)* rejected Hatusima's contention and transferred his taxon to, and placed their new species in, *Oxygyne*. However, *Ohashi (2015)* proposed to resurrect *Saionia* for the Japanese species of *Oxygyne,* stating that they differ from the Cameroon species in having (1) campanulate perianth, (2) bluish colour, (3) patent perianth lobes, (4) deflexed stamens of which the anthers are positioned below the base of the filament, (5) appendages on the style. Yet all these features, except the bluish colour of the perianth, are present in the second Cameroonian species described here as *O. duncanii*. This species appears closer in morphology to the Japanese species than it does to the geographically close type species, *O. triandra* and to the species of C.A.R., *O. frankei*. We maintain that flower colour alone is insufficient to support generic separation of the Japanese from the Cameroonian taxa. In the absence of molecular data, we are also reluctant to accept inclusion of *O. duncanii* in *Saionia,* even though it shares four of the five generic characters of *Saionia*.

Both *O. triandra* and *O. frankei* are known only from pressed specimens. Unlike all the other species of *Oxygyne,* there is no photograph or even a preserved field sketch of either species in the live state. We contend that several of the stated features of *O. triandra* (and the similar, newly described, *O. frankei*) that are given as points of generic separation may be artefacts of preservation, the state of anthesis or errors of artistic interpretation of the single known plant of *O. triandra*. For example, the first author has observed that in *O. duncanii* the perianth lobes in life are patent, but that on preservation in alcohol they tend to become erect, as in the drawing of *Oxygyne triandra*. Thus, it is entirely possible that this supposed generic distinction could also be an artefact of preservation. Similarly, the position of the anther in relation to the base of the declinate staminal filament can be interpreted from the drawing of *O. triandra* (and in the type specimen of *O. frankei*) as being at about the same level, which, while not "below the base of the filament" is very close to it. Furthermore, it is unlikely that the precise position of the anther in relation to the filament base would be reliably preserved in herbarium specimens, casting doubt on a second point of difference.

## Are there more *Oxygyne* species?

The possibility exists that further species of *Oxygyne* will yet be found. If so, they might be expected between SW Cameroon and Japan. However, the fact that numerous other achlorophyllous species in several genera have been discovered between these locations suggests that while possible, this is improbable. The reality is that additional species are most likely to be found in adjoining areas to those where the existing (or previously existing) species are found, namely in SW Region Cameroon, C.A.R. and in southern Japan. Yet the unexpected discovery of *O. frankei* in gallery forest of C.A.R. confounds the previous pattern, wherein *Oxygyne* species only occurred in high rainfall areas with high species diversity and other rare and endemic species. Certainly it would be worth exploring other gallery forest areas in tropical Africa for additional species. However, natural rainforest

habitat, including gallery forest, is rapidly being lost and degraded (M Cheek, pers. obs., 1994–2018). For this reason, effort is being invested to evidence and demarcate the highest priority areas for plant conservation as Important Plant Areas in the tropics *Darbyshire et al. (2017)*.

## ACKNOWLEDGEMENTS

Janis Shillito typed an earlier version of the manuscript. Eimear Nic Lughadha and two anonymous reviewers gave advice on an earlier version of the manuscript. James Byng first mentioned to the first author an unresolved thismiaceous specimen from C.A.R. George Gosline is thanked for independently subsequently uncovering the existence of that specimen, *Tisserant* 2623, and strongly advising that it be studied for the current paper, and for drawing Fig. 1. Dr Peguy Tchouto is thanked for helping to bring to light the first plant of *O. duncanii*. Josef Bogner encouraged the first author to publish on *Oxygyne*. The late Anacletus Koufani and Joseph Asonganyi (both formerly of YA), and Karen Sidwell (formerly of K and BM) are thanked for their efforts in searches for *Oxygyne duncanii* on 25th and 26th October 1992. Shigeo Yasuda (Yokohama) is thanked for his extensive and strenuous efforts in researching and translating the Japanese literature on *Oxygyne* with the assistance of Prof. Koyama (Makino Botanic Garden). Akiko Shimizu also kindly provided information on the literature of *Oxygyne* in Japan.

### Funding

Fieldwork funding leading to the discovery and collection of Oxygyne duncanii was received from the Overseas Development Administration of the UK government (now the Department for International Development). The funders had no role in study design, data collection and analysis, decision to publish, or preparation of the manuscript.

### Grant Disclosures

The following grant information was disclosed by the authors:
Overseas Development Administration of the UK government.

### Competing Interests

The authors declare there are no competing interests.

### Author Contributions

- Martin Cheek conceived and designed the experiments, performed the experiments, analyzed the data, contributed reagents/materials/analysis tools, prepared figures and/or tables, authored or reviewed drafts of the paper, approved the final draft.
- Hirokazu Tsukaya authored or reviewed drafts of the paper, approved the final draft, contributed descriptions of the Japanese species.
- Paula J. Rudall analyzed the data, contributed reagents/materials/analysis tools, prepared figures and/or tables, authored or reviewed drafts of the paper, approved the final draft, revised a draft of the paper; performed the anatomical work.

- Kenji Suetsugu prepared figures and/or tables, authored or reviewed drafts of the paper, approved the final draft, contributed photos of one species and data on the habitat, conservation status and protection of the Japanese species.

**Field Study Permissions**

The following information was supplied relating to field study approvals (i.e., approving body and any reference numbers):

The fieldwork was approved by the Institutional Review Board of the Royal Botanic Gardens, Kew entitled the Overseas Fieldwork Committee (OFC). This was issued under the terms of the five year Memorandum of Collaboration between Institute for Research in Agricultural Development (IRAD)-Herbier National du Cameroun and Royal Botanic Gardens, Kew, signed 5th Sept 2014.

**Data Availability**

All raw data used is included in the paper, e.g., the descriptions of new species.

**New Species Registration**

The following information was supplied regarding the registration of a newly described species:

*Oxygyne duncanii* LSID: 77178671-1

*Oxygyne frankei* LSID: 77178672-1.

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
