# Peer review of "Taxonomic monograph of Oxygyne (Thismiaceae), rare achlorophyllous mycoheterotrophs with strongly disjunct distribution"

_PeerJ, doi:10.7717/peerj.4828_

## Round 0.1 · original submission · Minor Revisions

When addressing this particular manuscript, I have taken account of the fact that the study of mycoheterotrophic plants is presently a highly competitive realm, even in its taxonomic aspects. I therefore wish to maximise the flexibility of response made available to the authors. Had the PeerJ electronic system permitted, I would therefore have ticked both the Minor Corrections and Major Corrections buttons, employing the following logic:

Both of the reviewers and I all agree that the present work merits publication and exceeds the minimum requirements of rigour established for PeerJ. We also concur with the major findings of the study. Given these over-arching conclusions, it may be that the authors wish to publish this work rapidly and painlessly, hence the Minor Revisions tick.

But if, alternatively, the Major Revisions button were to be ticked, the authors might consider taking a (very) little more time to convert their evidently thoroughly professional formal technical descriptions of these plants into a morphological matrix, also selecting a couple of outgroups from elsewhere in Burmanniaceae sensu lato. Both the compilation and analysis of such a matrix would require little effort, as the authors have, in practice, already performed most of the necessary work. Doing so would have the additional benefit of bringing the content of this study closer to the current title of the work (rather than de facto downgrading the title, as was suggested by Reviewer 1). Irrespective of their decision on this matter, I would ask the authors to redraft and include as a new Figure the relevant portion of the molecular tree published by Yokoyama et al. (2008), in order to provide a clearer context for comments made in the Introduction (e.g. Line 71). The authors may also wish to consider the suggestion of Reviewer 1 that they should expand the taxonomic treatment to make it more comprehensively monographic, though I suspect that the authors will wish to publish without the delay that would inevitably ensue.

I stand in permanent awe of taxonomists who can formally establish species on the back of between one and half a dozen herbarium specimens (e.g. Line 54); it signals a level of bravery that I personally have never achieved. But I would argue strongly that, where specimens are so thin on the ground, it behoves the authors to inform the reader the nature of those specimens. Are they dried? Pickled? Silica gel? How complete physically? These issues do not appear to be addressed directly until the Conclusions, and then only in passing (Line 851). I would also like to see all taxonomic studies begin with their own personalised short definition of a species concept, though my long-term attempts to persuade taxonomists of this need to do so have thus far fallen on exceptionally stony ground, and I will not abuse my role as handling editor by insisting on such an addition here.

I will, however, support the contention of Reviewer 1 that the manuscript should undergo a modest degree of restructuring, removing the RESULTS heading and combining the present Discovery and Extinction subsection and Conclusions into a more coherent DISCUSSION (perhaps also including some material transferred from the Introduction?). The Cheek-rich Bibliography appears thorough, though Reviewer 1 suggests one valid addition.

Most of the comments offered the reviewers and myself relate to presentation rather than content, and should be considered carefully in the context of a journal that offers the absolute minimum of copy editing. PeerJ is a WYSIWYG journal, so any pedantic authors wishing to avoid cringing at their own carelessness when the paper is published are well-advised to thoroughly self copy-edit. I would also point out that Reviewer 1 has kindly spent considerable time annotating his pdf copy fairly heavily with 'stickies', the majority of which are clearly helpful.

In this case, the paper is adequately formatted and well-written, only rarely lapsing into linguistic ambiguity (e.g. Line 412) or terminological carelessness (e.g. Line 633: 'predates' means eats, not antedates!) and containing laudably few typos. However, it scores less well in terms of consistency of punctuation: 'c.' and 'ca' are used interchangeably and randomly, as are hyphens versus n-dashes. Single spaces are often missing between value and measurement (e.g. 1mm), while spaces following full stops are often absent after 'c.' but archaic double spaces are used throughout the text at the end of sentences. Italicisation is occasionally absent where needed or alternatively overruns (e.g. caption to Fig. 1). And do the authors really wish to use a lower case 'x' where a multiplication symbol should lie? From my own experience, I would also caution them to consider carefully how best to present the dichotomous keys from a defensive viewpoint, as they are likely to find that the necessary formatting in lines has not been preserved in the ensuing proof.

Lastly, Reviewer 2 and I both expressed several (mostly minor) concerns about the submitted Figures. I do not believe that PeerJ can legitimately accept Figure 5 until it has been re-rendered by a hand possessing the artistic competence of that which drafted Figure 4. Figures 2 and 3 require the addition of narrow white strips to separate the four individual illustrations, and as noted by Reviewer 2, Figure 3 (top right) would benefit from enlargement (perhaps right-rotate and then crop accordingly?). Scales in Figure 3 are sufficiently ambiguous that a white bar should be added to at least one of the four illustrations, and although Figure 6 is of superb quality, that too presently lacks the necessary scale.

As I say, I do not believe that it will require great effort on the part of the authors to modestly improve this manuscript; I look forward to welcoming a revised version of this worthwhile contribution in short order.

Reviewer 1 ·

Basic reporting

NB, this reviewer’s comments are numbered consecutively.

1. This reviewer finds the present ms. to be well within the stated scope of the journal PeerJ. The ms. is also generally well written and structured in conformance with PeerJ standards (but see added comments below). The Introduction and Discussion sections are well researched, concise and have a logical ‘flow’, with good contact to relevant literature.

2. Below is a reference to one recently published article that seems relevant but may not have been seen by the present authors. They may want to cite it in their revised manuscript:

Sainge MN, Chuyong GB, Peterson, AT. 2017. Endemism and geographic distribution of African Thismiaceae. Plant Ecology and Evolution 150: 304-312. https://doi.org/10.5091/plecevo.2017.1196

3. On the ‘reviewing PDF’, this reviewer has added comments mainly of an editorial or copy-editing nature. These annotations should be considered part of this review and are intended to help the authors improve the quality of the final ms. for publication.

4. This reviewer has also checked the References list and finds an almost perfect, one-to-one correspondence between the entries on that list and the works cited in the main text. There are just two missing entries, both referring to web sites providing access to on-line databases:

• IPNI (continuously updated)
• Thiers et al. (continuously updated)

5. The ms. describes and names two new species, with methodology in conformance with PeerJ policy and fully meeting the criteria for valid publication in the International Code of Nomenclature for algae, fungi, and plants (McNeill et al., 2012).

6. This reviewer suggests a slightly different and more descriptive title: "Taxonomic revision of Oxygyne (Thismiaceae), a rare achlorophyllous mycoheterotroph with a remarkably disjunct distribution between tropical Africa and Japan". The terms ‘systematics’ and ‘taxonomy’ are certainly related, but systematics is broader and nowadays involves examining evidence of evolutionary relationships between organisms/taxa (usually with molecular data). The authors of the present ms. have reviewed the available literature on the systematics of Thismiaceae and Oxygyne, but they have not presented any new data or analyses along these lines (other than morphological comparisons). The work is fundamentally a taxonomic revision, and the title should be changed to reflect this.
According to PeerJ's editorial policy on publishing new species, it is also desirable for the title to mention the new species names (in order to increase visibility & discoverability of this information). This could be done, e.g., "Taxonomic revision of Oxygyne (Thismiaceae), with descriptions of O. duncanii and O. frankei, two new species from tropical Africa". The authors should decide if they want to implement this suggestion or stick with a title that emphasises the Africa/Japan disjunction.

7. Given that the manuscript is fundamentally a formal taxonomic treatment, the structure also needs to be modified somewhat. In particular, this reviewer recommends that the heading RESULTS should be removed and replaced with the heading TAXONOMIC TREATMENT. In the present version of the ms. there is no text under this heading, only an abrupt transition to the subheading Key to genera of Thismiaceae. It is therefore recommended that below the heading TAXONOMIC TREATMENT but before the subheading Key to genera of Thismiaceae, some basic information should be added about the family, similar to the following, with new corresponding entries added to the References list:

Thismiaceae J. Agardh, nom. cons. (Agardh, 1858)
Type genus: Thismia Griff. (Griffith, 1845)
Achlorophyllous mycoheterotrophs; for complete family description see Stevens (2001 onwards).
5 genera / approx. 55 species, mostly(?) rare and locally endemic; overall distribution widely scattered mostly in tropical and subtropical forests of South America, Africa and Southeast Asia; in Africa with highest concentration of species in Cameroon (Sainge et al. 2017).

8. In the RESULTS / TAXONOMIC treatment section, below the key to genera of Thismiaceae but before the beginning of the formal treatment of genus Oxygyne, the present ms. includes 7 short paragraphs of text under the subheading Discovery and extinction. This text seems out of place or even repetitive with information that is already presented in other parts of the ms., so the recommendation of this reviewer is for the subheading Discovery and extinction to be removed, with the accompanying text also deleted or moved / incorporated elsewhere in the ms. In the revised version, the start of the formal treatment of Oxygyne would thus immediately follow the key to genera of Thismiaceae.

9. After the formal treatment of Oxygyne, there are four paragraphs of text under the heading CONCLUSIONS. This text reads more like a discussion, leading this reviewer to recommend that the heading CONCLUSIONS should be removed and replaced with the heading DISCUSSION. If the revised ms. is to include a section called CONCLUSIONS then the text under this heading should be no longer than a single paragraph and should appear after the DISCUSSION. The text of the CONCLUSIONS should be limited to (1) a statement summarizing the paper’s main findings, (2) a more generalised statement about the importance of these findings, (3) perhaps an additional statement about unanswered questions or likely avenues for future work.

10. The figures in the present ms. are acceptable, but, in both their Abstract and the Introduction, the authors have clearly stated their intention for this work to be monographic. Given this intended scope, it is a shame there were no photographs or other illustrations provided for three of the six included species, i.e., Oxygyne triandra Schltr., O. shinzatoi (H.Ohashi) Tsukaya and O. hyodoi C. Abe & Akasawa. This reviewer would like to see photographs or line drawings added in revision so that all of the treated species have at least one illustration. In the case of O. triandra, the authors have also stated (line 737 of the present ms.) that the illustration in the protologue (Schlechter, 1906) was inaccurate in some respects, making a new illustration highly desirable. However, there are also some line drawings on the type sheet in B, and if the authors could secure permission to reproduce images of these drawings then these would be acceptable. Oxygyne shinzatoi and O. hyodoi also have previously published illustrations, and permissions could be sought from the original authors / copyright holders for these illustrations to be reproduced in the present work. Simply providing references to previously published illustrations is unacceptable for a work purportedly of monographic scope.

Experimental design

11. Materials and methods are perfectly fine for a work of this nature. The permits obtained by the first author to conduct field-work in Cameroon appear to be in good order.

Validity of the findings

12. This reviewer certainly agrees with the authors’ findings concerning the two new species described in the present work. The larger scope of the work (comprehensive revision of the entire genus) greatly enhances the value of these descriptions.

13. The most intriguing finding in the present work is that the newly described Cameroonian species Oxygyne duncanii appears closer morphologically to the three Japanese species than it does to O. triandra (of which the only known locality is about 12 km away from the type locality of O. duncanii). The authors have further noted (on lines 822824 of the present ms.) that if future molecular data support the distinction of Oxygyne and Saionia Hatus. as separate genera, then morphological comparisons indicate that O. triandra and Central African O. frankei would be placed together in Oxygyne, whilst O. duncanii and the three Japanese species would be placed together in Saionia. On lines 817821 they list four characters uniquely shared by O. triandra and O. frankei and seem to be of taxonomic importance: (1) erect very narrowly triangular to caudate perianth lobes; (2) dark brown lines (corresponding with the vascular supply) extending from the base of the perianth tube to the base of each perianth lobe; (3) reduced lamellae which are too small to occlude the perianth tube mouth; (4) a stigma with three subulate curved-erect lobes only, lacking lateral lobes. On the other hand, in their key to species of Oxygyne (starting on line 322), the leads of the first couplet have to do with geography and the colour of the flowers. While it may be true that the key should be viewed as nothing more than an aid to species identification, the fact remains there is a fundamental mismatch between the characters mentioned in the first couplet of the key on the one hand, and in the authors’ later discussion about characters of subgeneric or possibly generic importance. The authors need to address this mismatch somehow, either by fundamentally revising the key to Oxygyne species or by adding to the later discussion about subgeneric/generic characters a further note about the unnaturalness (artificiality) of the characters used in the key

Additional comments

No additional comments.

Annotated reviews are not available for download in order to protect the identity of reviewers who chose to remain anonymous.

·

Basic reporting

I consider that the manuscript meets all Peerj standards. The presentation of text is professional and authorative but at the same time very readable. It is very comprehensively referenced. Many primary references are obscure and might prove difficult to access, but this is inevitable when dealing with little-known taxa in remote localities

Experimental design

It is perfectly clear what the authors are aiming to achieve. For the sake of providing a detailed overview of all the species that the authors now consider to belong to Oxygyne, they have had to lean heavily on historical accounts as they had no first-hand access to living or pressed material of some taxa. This isn't ideal but I sense they have been as meticulous and objective as possible in drawing conclusions regarding taxa they have no experience of.

Validity of the findings

The conclusions are well stated and represent an important contribution to the taxonomy of a complicated and controversial group of heteromycotrophs. The authors concede that further generic ressignments are possible following molecular analyses, but given that at least two of the six taxa are extinct and others are proving so hard to locate, I can't see the molecular work being done anytine soon.

Additional comments

Despite this monograph dealing with a group of plants that are so little known and elusive that the vast majority of botanists are ever likely to encounter them, this monograph is very readable. It is interesting from a taxonomic perspective, intruiging from a phytogeographical perspective, and deeply depressing from a conservation perspective!

There is little to criticise. For the sake of completion, taxonomic decriptions of two of the Japanese species are reproduced verbatim from the monograph of Dr Tsukaya. They are substantial slabs of text, so I assume permission to do this was sought and obtained (if Dr Tsukaya is still alive, which I hope he is). I was surprised not to see him included in the acknowledgements. On a related note, the description of hyodoi is stated to be modified from Tsukaya, but I don't how this is possible given that none of the authors have seen the plant, alive or dead.

Given that the photos of duncani (Fig 4) were taken by the first author, there is scope to improve some of them. The two photos on the left of the plate can be closer views, because what is surrounding the plants is of no biological interest. Top right is basically a photograph of a penknife. I did eventually locate the in situ plant that is being referred to but if this image is to be retained at the same scale there need to be an arrow showing where it is. Bottom right is spoilt by sloppily allowing a hand lens to obscure most of the ruler included for scale.

The photos of root structure are fantastic but are they providing info on generic or sub-generic affiliations that wasn't present before?.

---

## Round 0.2 · accepted · Accept

I was asked to step in and make the final decision on this paper. I have read the reviews and the revisions and I am happy to accept your manuscript for publication. As pointed out by one of the reviewers, may I suggest that you consider replacing what are now Figures 8 and 10 with Dr Rudall's relabelled versions.

#